# Immunogenicity and safety of co-purified diphtheria, tetanus and acellular pertussis vaccine in 6-year-old Chinese children

Xuewen Tang[1,10], Yanhui Xiao[2,10], Jinhua Chen[3,10], Ying Su[4], Yang Zhou[1], Linyun Luo[2], Jiayou Zhang[5], Shaoxiang Liu[6], Rui Yan[1], Dewu Zhu[5], Wei Zhao[6], Yao Zhu[1], Xiao Ma[7], Yuli Jiang[8], Hailong Pan[2], Yuntao Zhang[2] & Hanqing He [1,9] ✉

In recent years, China has experienced a rapid increase in the number of pertussis cases among children aged 5 to 9. We conducted this phase 4 randomized, controlled trial (NCT05870631) in Zhejiang Province, China, to compare the immunogenicity and safety of co-purified diphtheria, tetanus, and acellular pertussis combined vaccine (co-purified DTaP) to diphtheria and tetanus combined vaccine (DT) in 6-year-old children. Between April 2023 to July 2023, 480 participants were randomized to receive one dose of co-purified DTaP ($n = 240$) or DT ($n = 240$). 28 days after vaccination, the seroconversion rates of the co-purified DTaP were 81.20% for anti-PT and 74.36% for anti-FHA, and achieved a significantly higher rate for anti-tetanus compared to the DT vaccine (97.01% vs. 87.05%, $P < 0.001$). Both vaccines showed 100% seroconversion for anti-diphtheria antibody. For all antibody types, the co-purified DTaP can induced higher geometric mean concentrations. More vaccine-related adverse events were reported with the co-purified DTaP. Most of the severity occurred in Grade 1 and 2. The booster dose of co-purified DTaP vaccination is safe and induces a significant anti-pertussis antibody response, it may be feasible to give a booster dose of co-purified DTaP instead of the recommended DT for 6-year-old children in China.

Pertussis, or whooping cough, is a highly contagious, vaccine-preventable infectious respiratory disease caused primarily by the Gram-negative bacterium *Bordetella* pertussis, which is spread mainly through droplets produced by coughing or sneezing[1]. It is characterized by paroxysms, inspiratory whoop, and post-tussive vomiting that typically lasts 4–8 weeks, it can cause severe illness and even death in infants under 3 months of age[2–4].

The resurgence of pertussis in recent years has remained a global public health challenge. After the removal of all COVID-19 mitigation measures, a sharp rise in pertussis cases has been reported in a number of countries or regions worldwide, including the United States[5,6], United Kingdom[7], France[8], Spain[9], Denmark[10], and South Korea[11]. Likewise, according to data released by the National Health Commission of China, between 1 January and 31 September 2024, 469,712 positive *Bordetella* pertussis cases were reported, which was more than 10 times the number of annual reports in 2023 (41,124 cases). The reasons for pertussis resurgence in the context of high vaccination coverage are complicated, including the waning of immunity induced

[1]Zhejiang Provincial Center for Disease Control and Prevention, Hangzhou, China. [2]China National Biotec Group Company Limited, Beijing, China. [3]Yiwu Center for Disease Control and Prevention, Jinhua, China. [4]Zhejiang Chinese Medical University, Hangzhou, China. [5]Wuhan Institute of Biological Products Company Limited, Wuhan, China. [6]Chengdu Institute of Biological Products Company Limited, Chengdu, China. [7]National Institutes of Food and Drug Control, Beijing, China. [8]Harvest integrated Research Organization (HiRO), Shanghai, China. [9]Zhejiang Key Lab of Vaccine, Infectious Disease Prevention and Control, Hangzhou, China. [10]These authors contributed equally: Xuewen Tang, Yanhui Xiao, Jinhua Chen. ✉e-mail: hanqinghe@cdc.zj.cn

by vaccines, the evolution of the pertussis pathogen and advances in detection sensitivity[12–14]. In addition, reduced exposure to *Bordetella pertussis* during lockdowns of the COVID-19 pandemic may have led to diminished natural boosting of immunity, contributing to the observed resurgence[15,16].

Pertussis vaccination is the most cost-effective strategy for preventing pertussis, reducing the risk of complications and severe cases[17,18]. Depending on the manufacturing technique, two types of acellular pertussis vaccines are available globally. One is the more widely used component- purified pertussis vaccine, which is mostly used in North America and Europe. Component-purified pertussis vaccine has clear contents, usually pertussis toxin (PT) and filamentous hemagglutinin (FHA) as a basis and pertactin (PRN) together with fimbriae (FIM) serotype 2 and serotype 3, making up the remaining ones[19]. Based on differences in target population and antigen content, component-purified pertussis vaccines that are more commonly used are the component-purified diphtheria, tetanus and acellular pertussis vaccine (component DTaP) for infants and young children, and the reduced antigen content diphtheria, tetanus, and acellular pertussis vaccine (Tdap) mainly for adolescents, adults, and pregnant women. The other co-purified pertussis vaccine, which is mostly used in Asian nations such as China and Japan. The components of the co-purified pertussis vaccine mainly include PT and FHA, and other *B. pertussis* protective antigens such as PRN, although the exact amount of each antigen in the vaccine is not clear[20]. In China, the most common pertussis vaccine is the diphtheria, tetanus and acellular co-purified pertussis combined vaccine (co-purified DTaP), which has been approved for use in children aged 3 months to 6 years. Before January 1, 2025, the China's National Immunization Program (NIP) recommending a four-dose of co-purified DTaP schedule at 3, 4, 5, and 18 months. In addition to co-purified DTaP, there are two non-NIP DTaP vaccines (i.e., private, out-of-pocket expense) available for infants and toddlers in China: a pentavalent component DTaP-IPV/Hib combination vaccine comprising component purified pertussis vaccine, and a quadrivalent DTaP-Hib combination vaccine comprising co-purified pertussis vaccine. In certain countries, such as the United States, Canada and South Korea, after completing four doses of pertussis vaccine for children under the age of two, it is recommended that children aged 4–6 receive a booster dose of component DTaP or Tdap, and that adolescents, pregnant women, and adults also receive a booster dose of Tdap[21–24]. However, no new component pertussis vaccine (component DTaP or Tdap) has been licensed for use in China in children aged 6 and above, adolescents, adults, or pregnant women.

Recent epidemiological studies of pertussis in China have revealed a significant rise in the frequency of pertussis cases in children over the age of 6 years[25–29], indicating that preschool children should be immunized with one dose of pertussis vaccine. In the absence of new pertussis vaccines, it is critical and essential to explore a booster immunization strategy for 6-year-old Chinese children based on co-purified DTaP. Therefore, we conducted a phase 4 trial to assess the immunogenicity and safety of co-purified DTaP in 6-year-old children in Zhejiang province, China.

## Results

### Trial population

Between April 2023 and July 2023, 527 participants were assessed for eligibility, and 480 participants randomly assigned to either the co-purified DTaP or diphtheria and tetanus combined vaccine (DT) arms. Of these, 480 participants were included in safety set (SS) for safety and 458 in per-protocol set (PPS) for immunogenicity. A total of 18 participants (3.75%) were withdrawn from the trial (Fig. 1).

Table 1 shows the demographic and baseline characteristics information. In the SS population, the mean age was 73.49 months in the co-purified DTaP and 73.43 months in the DT. There were 128 (53.33%) female participants in the co-purified DTaP and 125 (52.08%) female participants in the DT. All participants were of Asian race, with 477 (> 99%) being of Han Chinese ethnicity. Demographic and baseline characteristics of the participants were generally well balanced between the two arms.

### Immunogenicity

Before vaccination, the seropositive rates against pertussis, diphtheria, and tetanus in the co-purified DTaP and DT varied between 3.57% and 91.45%, with geometric mean concentrations (GMCs) ranging from 0.06 to 10.89. The seropositive rate of anti-PT antibody was slightly higher in the co-purified DTaP than the DT (8.97% vs. 3.57%, *P* = 0.018). There was no significant difference in seropositive rates for other antibodies between the two arms. The baseline antibody GMCs or median concentrations against pertussis, diphtheria, and tetanus were similar between the two arms (Table 2 and Fig. 2).

On 28 days after vaccination, the seropositive rates of anti-pertussis antibodies in the co-purified DTaP increased to 85.47% (200/234, 95% CI: 80.38–89.41) for PT and 97.44% (228/234, 95% CI: 94.52-98.82) for FHA. The seroconversion rate for anti-PT antibody was 81.20% (190/234, 95% CI: 75.70–85.68), with the lower limit of the 95% CI exceeding 70% (75.70% > 70%). The seroconversion rate for anti-FHA antibody was 74.36% (174/234, 95% CI: 68.40–79.53). For anti-diphtheria (anti-DT) antibody, the seropositive rates and seroconversion rates reached 100% in both arms. For anti-tetanus (anti-TT) antibody, the seropositive rates were 100% in both arms, but the seroconversion rates were higher in the co-purified DTaP than in the DT (97.01% [227/234, 95% CI: 93.96–98.54] vs. 87.05% [195/224, 95% CI: 82.03–90.83], *P* < 0.001). The anti-DT and anti-TT had higher GMCs or median concentrations in the co-purified DTaP than in the DT (Table 2 and Fig. 2). Participants with negative pertussis antibodies at baseline had a higher seroconversion rate than those with positive pertussis antibodies at baseline in the co-purified DTaP (Supplementary Table 1).

### Safety

Among all participants, more vaccine-related adverse events were reported within 28 days of any immunization in the co-purified DTaP than the DT (23.75% [57/240] vs. 9.17% [57/240], *P* < 0.001). The incidence of solicited local adverse events (AEs) in the co-purified DTaP was higher than that in the DT (21.67% [52/240] vs. 7.08% [17/240], *P* < 0.001), with swelling and redness being the most common solicited symptoms. Solicited systemic AEs were reported by 5.00% (12/240) in the co-purified DTaP and 2.50% (6/240) in the DT. There was no significant difference in solicited systemic AEs between the two arms. The most common solicited symptoms were fever (Table 3).

All vaccines administered during the trial period were well-tolerated. The severity of vaccine-related adverse events was mainly Grade 1 and 2. There was no significant difference in the incidence of ≥Grade 3 vaccine-related adverse events between the two arms (0.83% [2/240] vs. 0% [0/240], *P* = 0.499) (Table 4). Two participants, both in the co-purified DTaP, occurred ≥ Grade 3 vaccine-related adverse events within 3 days after immunization, including one participant with fever and one participant with injection-site swelling or redness. There were no deaths among any participants during the study period. None of the vaccine-related serious adverse event (SAE) occurred at any time during the study period. A 6-year-old boy developed fever three days after co-purified DTaP vaccination and was hospitalized for a diagnosis of Miller-Fisher syndrome (MFS) that as a variant of Guillain–Barré syndrome (GBS), due to limb weakness and binocular strabismus. This SAE was reported in the co-purified DTaP, which was considered unrelated to the vaccine.

## Discussion

With the resurgence of pertussis in China, it seems urgent and necessary to introduce one dose of pertussis vaccine booster immunization for preschool children. A study based in Shanghai, China, evaluated the

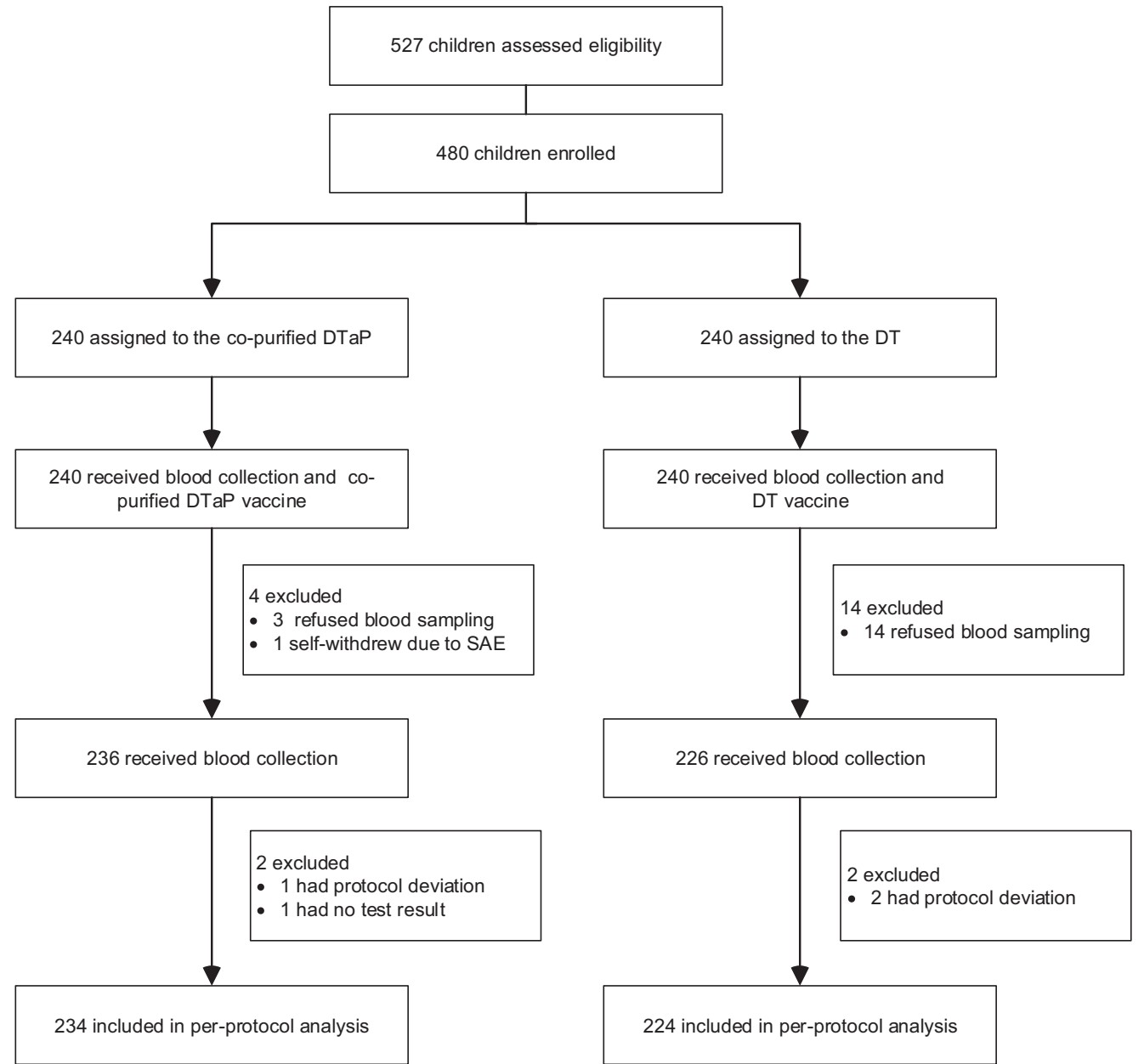

**Fig. 1 | Trial profile.** co-purified DTaP = co-purified diphtheria, tetanus and acellular pertussis combined vaccine. DT = Diphtheria and tetanus combined vaccine.

cost-effectiveness analysis of a booster dose of the co-purified DTaP in preschool children using a Markov decision tree model and concluded that it would be a cost-saving vaccination strategy[30]. In our trial conducted in Zhejiang province, we found that after 6-year-old children were vaccinated with the co-purified DTaP, the seroconversion rate of anti-PT reached 81.20% (target value:70%), and both seropositive rates and antibody GMCs against pertussis (anti-PT and anti-FHA) increased significantly. Furthermore, higher levels of antibodies against diphtheria and tetanus were obtained with the co-purified DTaP. The safety profile of the co-purified DTaP in children aged 6 years is acceptable, with AEs characterized by predominantly mild or moderate severity of symptoms at the injection site.

At present, there are three kinds of pertussis vaccines in use worldwide, including whole-cell pertussis vaccines, component-purified acellular pertussis vaccines and co-purified acellular pertussis vaccines[17]. Since co-purified DTaP is only used in a few countries, and there is no component DTaP available for 6-year-olds in China, our trial was the first to evaluate the immunogenicity and safety of co-purified

DTaP in 6-year-old children. Limited research has compared the immunogenicity of co-purified DTaP vaccines with that of component DTaP vaccines, the results show that the pertussis antibodies induced by component DTaP vaccines are superior to those of the co-purified DTaP vaccine. A study evaluated the immunogenicity of four doses of component DTaP-IPV/Hib and co-purified DTaP in Chinese children and found that the initial GMCs against pertussis after vaccination of component DTaP-IPV/Hib was higher than that of co-purified DTaP, declined to similar levels at 72 months after immunization[31]. Similar to many industrialized countries with high vaccination coverage, China has also reported a shift in the circulating strains of *Bordetella* pertussis in recent years[32–35]. A significant divergence between circulating strains ptxP3 and co-purified DTaP vaccine strain (CS strain: ptxP-1/prn-1/ptxA-2/ffm2-1/ffm3-1/tcfA2) was observed in China[35,36]. The shift of ptxP3 variants from ptxP1 vaccine strains may facilitate vaccine-elicited immunity escape, which could partially explain the resurgence of pertussis. The correlation between pertussis antibody thresholds and efficacy and durable protection remains uncertain[37]. Although the

**Table 1 | Demographic and baseline characteristics**

| | co-purified DTaP (N = 240) | DT (N = 240) |
|---|---|---|
| **Age (months), mean (SD)** | 73.49 (1.95) | 73.43 (1.88) |
| **Sex, n (%)** | | |
| Male | 128 (53.33%) | 125 (52.08%) |
| Female | 112 (46.67%) | 115 (47.92%) |
| **Ethnic, n (%)** | | |
| Han | 239 (99.58%) | 238 (99.17%) |
| other | 1 (0.42%) | 2 (0.83%) |
| **Height (cm), mean (SD)** | 117.71 (4.82) | 118.07 (5.49) |
| **Weight (kg), mean (SD)** | 21.66 (3.78) | 21.85 (3.64) |
| **BMI (kg/m²), mean (SD)** | 15.58 (2.19) | 15.64 (2.19) |

Data are mean (SD) or n (%). n = number of participants. % = proportion of participants. SD = Standard deviation; BMI = body-mass index; co-purified DTaP, co-purified diphtheria, tetanus and acellular pertussis combined vaccine; DT, diphtheria and tetanus combined vaccine.

seroconversion rate against PT in our trial met the pre-specified target, it is necessary to evaluate the effectiveness of the co-purified DTaP vaccine against the current circulating pertussis strains.

Several studies have reported similar immunogenicity and safety results with the fifth dose of a component DTaP vaccine in 6-year-old children, however, the vaccine, study population, and definition of seroconversion of pertussis antibody vary between these studies. A study conducted in the United States involving 200 children aged 4–6 years found that the seroconversion rates of anti-pertussis antibodies (anti-PT, anti-FHA, and anti-PRN) after one dose of 3-component DTaP were above 90%, with injection site pain, induration, and swelling being the most common AEs[38]. Another study reported that after booster vaccination with six different component DTaP in children aged 4–6 years resulted in 4-fold increase rates of anti-PT antibody ranging from 70% to 100%, and 4-fold increase rates of anti-FHA antibody ranging from 50% to 100%, with injection site pain and swelling being common AEs[39]. The aim of this study was to evaluate the immunogenicity and safety of co-purified DTaP vaccination in health children at age 6 years, and the vaccine effectiveness, immunogenicity and safety of a booster dose at age 6 years needs to be further explored in larger real-world populations (including special health populations) in China.

Several previous studies have reported the results on the immunogenicity and persistence of the co-purified DTaP produced by Chengdu Institute of Biological Products, conducted in Jiangsu, Anhui, and Sichuan provinces in China. At 3, 4, and 5 months of age, after three doses, seroconversion rates of antibodies against PT and FHA were above 90% and 100% for tetanus and diphtheria respectively[40]. At the age of 18 months, the seropositive rates of antibodies against pertussis (PT and FHA) had decreased to less than 10%, while the seropositive rates of antibodies against diphtheria and tetanus remained at a high level (above 98%). After booster vaccination with one dose of co-purified DTaP, the seropositive rates of antibodies against pertussis, diphtheria and tetanus were all above 98%[41]. According to the data of our trial, it was found that 6-year-olds already had low pertussis antibody seropositive rates before vaccination, with less than 10% against PT and 30% against FHA, and a high seropositive rate of tetanus antibodies before booster vaccination (> 90%). A study of 6- to 8-year-old children in China who had completed 4 doses of co-purified diphtheria, tetanus, and whole cell pertussis combined vaccine (DTwP) showed the following pre-booster seropositivity rates: 5.6% for anti-PT, 40.1% for anti-FHA, 72.2% for anti-DT, and 83.3% for anti-TT, also demonstrating high variability in seropositivity rates between different antigens (5%–83%)[42]. All children enrolled in this trial were needed to have received four doses of the DTaP vaccination and self-reported no

history of illness with related diseases. As a result, the reason for the high variation of seropositive rates for three antibodies among 6-year-old children might be attributed to different antigen characteristics and the detection sensitivity of vaccines. This significant variability suggests that a pertussis vaccine booster is needed at age 6 years, the interval between booster vaccinations of tetanus vaccine should be considered when using co-purified DTaP vaccine as a booster vaccine, or a new monovalent pertussis vaccine should be developed as an alternative. Participants in our trial who received co-purified DTaP had higher seroconversion rates of pertussis antibody than those who received DT, regardless of baseline pertussis antibody levels before booster vaccination. Some studies on Tdap booster immunization have shown that participants with high baseline pertussis antibody levels have a higher antibody response one month after vaccination, suggesting that the long-lasting memory formed by previous exposure or vaccination history may affect the vaccine response[43]. However, participants who were positive for pertussis antibodies at baseline had lower seroconversion rates of PT and FHA after a booster vaccination with co-purified DTaP than those who were negative for pertussis antibodies at baseline in our trial, which could be attributed to the different definitions of seroconversion rates for pre-immunization positive or negative.

Our trial found that the incidence of vaccine-related AEs reported in the co-purified DTaP was higher than that in the DT within 28 days after vaccination, and the AEs of co-purified DTaP immunization in 6-year-old children were mostly mild to moderate injection symptoms such as redness and swelling, which were well tolerated. A study conducted in China in 2007 showed that the safety data of the Tdap vaccine, which was briefly approved in China for children aged 6 to 8, was comparable to that of the DT vaccine among 6-year-old children[42]. In our trial, more than 15% of the participants who received co-purified DTaP experienced redness or swelling. A large number of studies in the United States, Canada and other countries have found that local reactions are the most common AEs after Tdap or component DTaP booster vaccination in similar age groups[44,45], which is consistent with the safety results of our study. According to the adverse event following immunization (AEFI) passive surveillance data released by the Chinese Center for Disease Control and Prevention during 2021-2022, the incidences of reported AEFIs were 80.58 cases per 100,000 doses for co-purified DTaP, 49.71 cases per 100,000 doses for DT[46]. The incidences of reported AEFIs for co-purified DTaP was higher than that of DT after large-scale use in China, which was consistent with our safety findings. However, throughout the study period, there was one SAE of MFS reported in the co-purified DTaP, which was considered unrelated to the vaccine. There were no further occurrences of MFS in the AEFI system after the administration of this lot of co-purified DTaP (lot number: 20220311) produced by Chengdu Institute of Biological Products. MFS, a variant of the GBS, is an extremely rare disease with an annual incidence of about one patient per one million population[47]. Since the co-purified DTaP was approved for large-scale use in China in the 1990s, only rare cases of GBS after vaccination has been reported, but the causal relationship has yet to be confirmed[48]. A study that used the Vaccine Safety Datalink (VSD) to analyze the association between Tdap vaccination and GBS in subjects aged 10-64 years from 2005 to 2009 found that Tdap did not increase the risk of GBS within 6 weeks of vaccination[49,50]. The SAE in our trial might have been a coincidence event, but it suggests that the risk of GBS needs to be monitored if co-purified DTaP is widely used among 6-year-old children in China.

Our study has some potential limitations. First, although guardians were asked to self-report a history of pertussis at enrollment, active surveillance for pertussis-like symptoms was not conducted in participants or their close contacts prior to enrollment, and participants were not screened for pertussis infection based on a

**Table 2 | Immune response to co-purified DTaP or DT before and day 28 after vaccination in the per-protocol population**

| Antibody | Timing | Seropositive/ Seroconversion | | | | | GMC | | | | |
|---|---|---|---|---|---|---|---|---|---|---|---|
| | | co-purified DTaP (N=234) | | DT (N=224) | | P | co-purified DTaP (N=234) | | DT (N=224) | | P |
| | | n (%) | 95% CI | n (%) | 95% CI | | value — 95% CI | | value — 95% CI | | |
| **anti-PT** | | | | | | | | | | | |
| Seropositive | Pre | 21(8.97) | 5.94–13.33 | 8(3.57) | 1.82–6.89 | 0.018 | 2.06 — 1.69–2.53 | | 1.67 — 1.40–2.00 | | 0.123 |
| Seropositive | Post | 200(85.47) | 80.38–89.41 | 12(5.36) | 3.09–9.13 | | 51.95 — 46.30–58.30 | | 1.97 — 1.65–2.34 | | |
| Seroconversion | Post | 190(81.20) | 75.70–85.68 | 5(2.23) | 0.96–5.12 | | – — – | | – — – | | – |
| **anti-FHA** | | | | | | | | | | | |
| Seropositive | Pre | 85(36.32) | 30.43–42.66 | 69(30.80) | 25.12–37.13 | 0.211 | 10.89 — 8.78–13.50 | | 9.84 — 8.06–12.02 | | 0.501 |
| Seropositive | Post | 228(97.44) | 94.52–98.82 | 75(33.48) | 27.63–39.89 | | 116.75 — 105.04–129.76 | | 11.03 — 9.21–13.20 | | |
| Seroconversion | Post | 174(74.36) | 68.40–79.53 | 8(3.57) | 1.82–6.89 | | – — – | | – — – | | – |
| **anti-DT** | | | | | | | | | | | |
| Seropositive | Pre | 87(37.18) | 31.24–43.53 | 77(34.38) | 28.47–40.81 | 0.531 | 0.06 — 0.05–0.07 | | 0.06 — 0.05–0.07 | | 0.586 |
| Seropositive | Post | 234(100.00) | 98.38–100.00 | 224(100.00) | 98.31–100.00 | 1.000 | 4.08 — 3.59–4.64 | | 2.98 — 2.67–3.33 | | < 0.001 |
| Seroconversion | Post | 234(100.00) | 98.38–100.00 | 224(100.00) | 98.31–100.00 | 1.000 | – — – | | – — – | | – |
| **anti-TT** | | | | | | | | | | | |
| Seropositive | Pre | 214(91.45) | 87.17–94.40 | 203(90.63) | 86.09–93.79 | 0.756 | 0.38 — 0.34–0.44 | | 0.35 — 0.31–0.41 | | 0.388 |
| Seropositive | Post | 234(100.00) | 98.38–100.00 | 224(100.00) | 98.31–100.00 | 1.000 | 11.63 — 10.75–12.60 | | 3.45 — 3.17–3.75 | | < 0.001 |
| Seroconversion | Post | 227(97.01) | 93.96–98.54 | 195(87.05) | 82.03–90.83 | < 0.001 | – — – | | – — – | | – |

GMC = geometric mean concentrations. CI = confidence intervals. A single-arm objective performance criteria for anti-PT antibody seroconversion rates against a 70% target value, where the criterion for success requires the lower limit of the 95% CI for the anti-PT seroconversion rate to exceed 70%. The two-sided $\chi 2$ test to compare seropositive rates and seroconversion rates between arms. The Wilson (score) method was used to calculate 95% CIs for seropositive rates and seroconversion rates. The GMCs were compared between arms using the two-sided Student's t test. anti-PT, anti-pertussis toxin antibody; anti-FHA, anti-filamentous hemagglutinin antibody; anti-DT, anti-diphtheria antibody; anti-TT, anti-tetanus antibody; co-purified DTaP, co-purified diphtheria, tetanus and acellular pertussis combined vaccine; DT, diphtheria and tetanus combined vaccine.

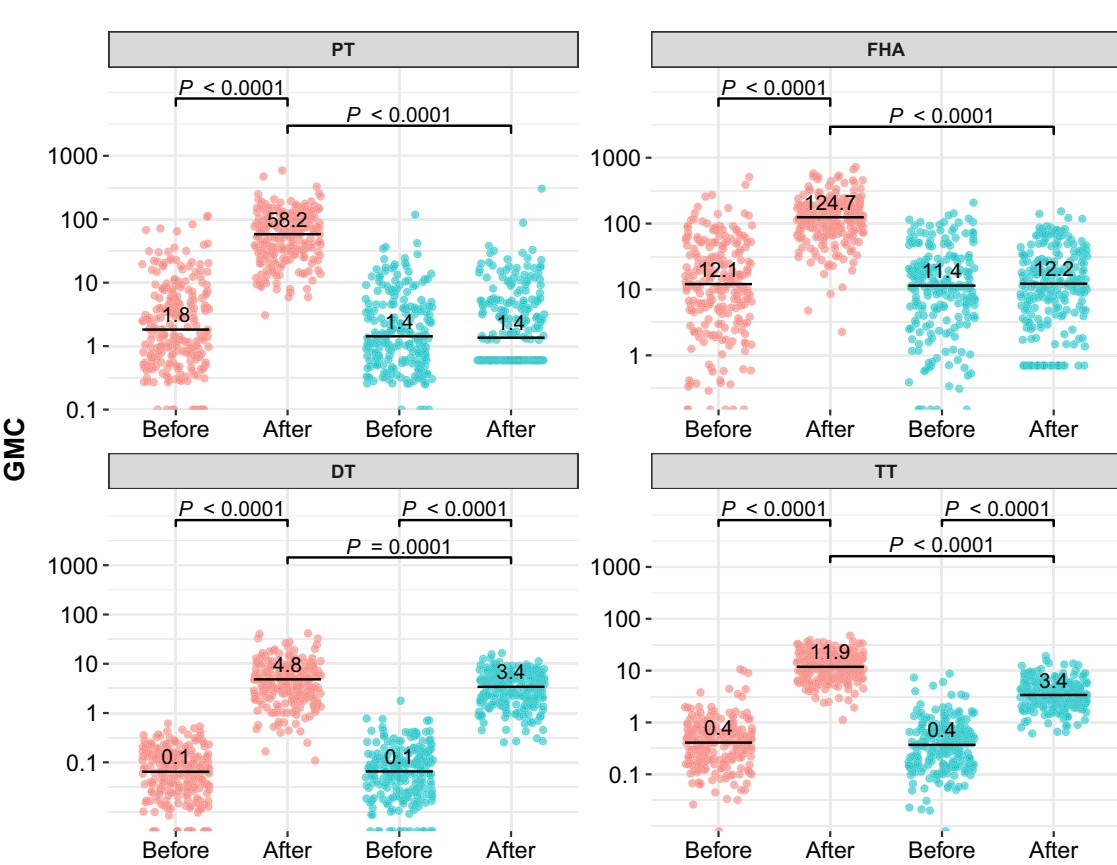

**Fig. 2 | Comparison of median concentrations after booster with co-purified DTaP or booster with DT.** Antibody concentrations against PT, FHA, DT and TT of participants boosted with the co-purified DTaP (*n* = 234) or DT (*n* = 224) before booster vaccination or 28 days after booster vaccination. Each point represents the antibody concentration of the participants in the PPS dataset. The median concentrations are shown as horizontal, respectively. The two-sided Wilcox rank-sum test was used for statistical analysis. *P*-values are shown above the data compared.

standardized case definition (such as laboratory confirmation, clinical criteria, and/or epidemiological links) before vaccination. As a result, some participants with asymptomatic or moderate symptomatic pertussis infection who did not visit health care facilities may have been recruited in the trial, which could have confounded the immunogenicity results. Second, we conducted our trial in Zhejiang Province, which has a high reporting rate of pertussis cases in China. This might impose a potential constraint on the generalizability of the findings to populations with differing background rates of pertussis circulation, natural immunity, and potentially heterogeneous circulating *Bordetella* pertussis strains. Finally, our trial used a block randomization with a block size of 8 rather than a random block size, which increased the predictability. Nevertheless, the results showed that the baseline characteristics of the participants were comparable between the two arms.

In recent years, a substantial increase in pertussis incidence was observed in China, with a pronounced surge among children aged 5–9 years, while severe and fatal cases remain largely concentrated among unvaccinated infants aged ≤ 3 months. To enhance the immune protection for school-aged children and infants aged ≤ 3 months, China has implemented a new immunization schedule containing pertussis vaccine from January 1, 2025, which changes from one dose of co-purified DTaP vaccine at 3, 4, 5, and 18 months of age to one dose at 2, 4, 6, 18 months, and 6 years of age[51]. Our study found that in 6-year-old children, co-purified DTaP vaccination achieved a 70% seroconversion rate for pertussis anti-PT antibodies, significantly increased levels of pertussis anti-PT and anti-FHA antibodies, did not reduce diphtheria

or tetanus antibody levels compared with DT vaccination, and demonstrated a good safety profile. These results support the strategy of administering a fifth dose of co-purified DTaP vaccine instead of the DT vaccine to 6-year-old children in China's NIP. The data on the use of co-purified DTaP in this study can also provide a reference for countries or regions where pertussis resurgence has occurred, especially those with an increasing proportion of cases among school-aged children, to adjust their immunization strategies.

## Methods
### Ethics statement
This trial complies with all relevant ethical regulations, and the protocol, informed consent, and amendments was approved by the Ethics Committee of Zhejiang Provincial Center for Disease Control and Prevention (2023-008-1). This trial is conducted in accordance with the Declaration of Helsinki and Good Clinical Practice guidelines. Informed consent, signed by a parent or legal guardian, was obtained from all participants before any study procedure.

### Study design
To compare the immunogenicity and safety of the co-purified DTaP with the DT recommended by China's NIP in 6-year-old children, a randomized, controlled, open-label, phase 4 trial was conducted to control potential confounding factors that might affect the results. From April to July 2023, our trial was conducted in the Fuyang District of Hangzhou City and Tongxiang City of Zhejiang Province, China. This trial was registered with ClinicalTrials.gov (NCT05870631).

**Table 3 | Reported vaccine-related adverse events following any vaccination within 28 days**

| Adverse events | co-purified DTaP (N = 240) | DT (N = 240) | P |
|---|---|---|---|
| **Total, n (%)** | 57 (23.75) | 22 (9.17) | < 0.001 |
| **Solicited systemic, n (%)** | 12 (5.00) | 6 (2.50) | 0.149 |
| Fever, n (%) | 10 (4.17) | 6 (2.50) | 0.309 |
| Fatigue/weakness, n (%) | 2 (0.83) | 6 (2.50) | 0.499 |
| Decreased appetite, n (%) | 0 (0) | 0 (0) | – |
| Nausea, n (%) | 0 (0) | 0 (0) | – |
| Vomiting, n (%) | 2 (0.83) | 0 (0) | 0.499 |
| Diarrhea, n (%) | 0 (0) | 0 (0) | – |
| Allergy, n (%) | 0 (0) | 0 (0) | – |
| Myalgia, n (%) | 0 (0) | 0 (0) | – |
| Arthralgia, n (%) | 0 (0) | 0 (0) | – |
| **Solicited local, n (%)** | 52 (21.67) | 17 (7.08) | < 0.001 |
| Pain, n (%) | 17 (7.08) | 10 (4.17) | 0.166 |
| Induration, n (%) | 15 (6.25) | 11 (4.58) | 0.420 |
| Swelling, n (%) | 38 (15.83) | 10 (4.17) | < 0.001 |
| Rash, n (%) | 1 (0.42) | 1 (0.42) | > 0.999 |
| Redness, n (%) | 40 (16.67) | 7 (2.92) | < 0.001 |
| Pruritus, n (%) | 15 (6.25) | 3 (1.25) | 0.004 |
| **Non-solicited, n (%)** | 3 (1.25) | 1 (0.42) | 0.623 |

Adverse events were compared between arms using the two-sided χ2 test or Fisher's exact test. co-purified DTaP, co-purified diphtheria, tetanus and acellular pertussis combined vaccine; DT, diphtheria and tetanus combined vaccine.

**Table 4 | Severity of vaccine-related adverse events reported within 28 days**

| Adverse events | co-purified DTaP (N = 240) | DT (N = 240) | P |
|---|---|---|---|
| **Total, n (%)** | | | |
| Grade 1 | 49 (20.42) | 18 (7.50) | < 0.001 |
| Grade 2 | 41 (17.08) | 10 (4.17) | < 0.001 |
| ≥ Grade 3 | 2 (0.83) | 0 (0) | 0.499 |
| **Systemic, n (%)** | | | |
| Grade 1 | 6 (2.50) | 1 (0.42) | 0.122 |
| Grade 2 | 8 (3.33) | 5 (2.08) | 0.399 |
| ≥ Grade 3 | 1 (0.42) | 0 (0) | > 0.999 |
| **Local, n (%)** | | | |
| Grade 1 | 45 (18.75) | 16 (6.67) | < 0.001 |
| Grade 2 | 36 (15.00) | 5 (2.08) | < 0.001 |
| ≥ Grade 3 | 1 (0.42) | 0 (0) | > 0.999 |

Adverse events were compared between arms using the two-sided χ2 test or Fisher's exact test. co-purified DTaP, co-purified diphtheria, tetanus and acellular pertussis combined vaccine; DT, diphtheria and tetanus combined vaccine.

## Participants

Participants were eligible for inclusion if they met the following criteria: aged between 6 and 7 years; ≥ 14 days since the last vaccination; received 4 doses of diphtheria, tetanus, and pertussis vaccine; not vaccinated with diphtheria, tetanus, and pertussis immunization products in the past 3 years; had no history of pertussis, diphtheria, or tetanus; and had a body temperature of ≤ 37.3 °C. The exclusion criteria included a history or family history of allergies, convulsions, epilepsy, encephalopathy, or psychosis; a history of severe allergies after vaccination; with immunodeficiency, cancer treatment, treatment with immunosuppressive agents (oral steroids); injection of non-specific immunoglobulins within one month before enrollment; a history of thrombocytopenia or other coagulation disorders.

## Randomization and masking

Participants were randomly (1:1) assigned to receive either co-purified DTaP (co-purified DTaP arm) or DT (DT arm). Randomization was performed using a block randomization method with a block size of 8, and random number tables were generated by a statistician using SAS 9.4. Random numbers were placed in envelopes, and each eligible participant was assigned a random number, which was used to identify all procedures to be carried out. Participants and investigators were not blinded to the group assignment, but laboratory technicians were blinded to the vaccines received in each arm.

## Vaccines

Co-purified DTaP was produced by Chengdu Institute of Biological Products (0.5 ml, lot number: 20220311, Chengdu, China), and contained at least 4.0 IU of acellular pertussis toxoid, 30 IU of diphtheria toxoid, and 40 IU of tetanus toxoid. The general principles of the co-purification process: The *Bordetella* pertussis vaccine strain was cultured via fermentation to generate a bacterial harvest containing key pertussis antigenic components: PT, FHA, PRN, and FIM2/3, etc. Following co-purification via ammonium sulfate precipitation, extraction, and sucrose density gradient centrifugation, the pertussis bacterial harvest was purified to yield a refined antigen solution containing multiple target immunogens. Chemical detoxification with glutaraldehyde generated the pertussis bulk solution, which was used to prepare the co-purified DTaP. The Chinese Pharmacopoeia stipulates that for co-purified DTaP vaccines, the content of PT and FHA shall constitute no less than 85% of the total protein content. Co-purified DTaP was approved for use in children aged of 3 months to 6 years. The manufacturing technique and formulation of the co-purified DTaP for 6-year-olds in this trial are the same as China's NIP vaccine used for the primary three-dose series at 3, 4, and 5 months old, as well as booster immunization at 18 months.

DT was manufactured by Wuhan Institute of Biological Products (0.5 ml, lot number: 20211202020-2, Wuhan, China) and contained at least 30 IU of diphtheria toxoid, 40 IU of tetanus toxoid. DT was approved for use in children under the age of 12, and China's NIP recommended one dose of the DT at the age of 6 by 2025. The two types of vaccines belong to NIP vaccines in China and were purchased by Zhejiang Provincial Center for Disease Control and Prevention for use in this trial.

## Procedures

After enrollment, baseline demographic data were collected for each participant through a uniform questionnaire, which included date of birth, sex, height, weight, and immunization history. On day 0, the co-purified DTaP arm was given one dose of co-purified DTaP, and the DT arm received one dose of DT.

Blood samples were collected on day 0 before vaccination and day 28 after vaccination. Blood sample collection after vaccination can be completed within 28–42 days. Concentrations of antibodies against pertussis, diphtheria, and tetanus were measured by the National Institutes of Food and Drug Control using the Enzyme-Linked Immunosorbent Assay (ELISA) method. Seropositive was defined by anti-pertussis component antibodies (anti-PT and anti-FHA) concentration of ≥ 20 IU/ml, anti-DT and anti-TT antibody concentrations of ≥ 0.1 IU/ml. Seroconversion of anti-pertussis component antibodies was defined as anti-pertussis antibody concentrations < 20 IU/ml before vaccination and ≥ 20 IU/ml after vaccination, or antibody concentrations ≥ 20 IU/ml before vaccination and increased ≥ 4 times after vaccination. Seroconversion of anti-DT and anti-TT was

defined as antibody concentrations < 0.1 IU/ml before vaccination and ≥ 0.1 IU/ml after vaccination, or if antibody concentrations were seropositive before vaccination and increased ≥ 4 times after vaccination.

Any local or systemic AEs occurring between 0 and 28 days after vaccination were recorded, and the investigators collected AEs or SAEs within 6 months of vaccination using the Chinese AEFI system. The solicited local AEs included pain, induration, swelling, rash, redness, and pruritus. The solicited systemic AEs included fever, fatigue/weakness, decreased appetite, nausea, vomiting, diarrhea, allergy, myalgia, and arthralgia. Solicited AEs occurring within 7 days of vaccination were considered vaccine-related. Unsolicited AEs from day 0 to day 28 after vaccination and solicited AEs from day 8 to day 28 days after vaccination were assessed as relevant by the investigators. The severity of AEs was assessed in accordance with the Guidelines for Classification of Adverse Events in Clinical Trials of Preventive Vaccines (2019 edition) by the National Medical Products Administration.

## Outcomes
The primary outcomes of the study were the seroconversion rates and seropositive rates of anti-pertussis antibodies in the co-purified DTaP, as well as a comparison of seroconversion rates and seropositive rates of anti-DT and anti-TT antibodies in the co-purified DTaP and DT. The secondary outcome was the rates of AEs 28 days after immunization with the co-purified DTaP or DT, as well as antibody GMCs against pertussis, diphtheria, and tetanus before and after vaccination.

## Statistical analysis
As PT is a key protective antigen in pertussis vaccines[52], the seroconversion rate for anti-PT antibodies was used for sample size estimation. The sample size was estimated using an objective performance criteria method. Assuming the target value was 70%[42] of the seroconversion rate of anti-PT in the co-purified DTaP after vaccination, with a two-sided α of 0.05, a power of 90%, and a potential loss to follow-up of 15%, we estimated at least a sample size of 240 participants per arm.

Baseline characteristics were reported using means (SD) for continuous variables and n (%) for categorical variables. Immunogenicity was assessed in the PPS, which included all participants who met the inclusion/exclusion criteria and completed vaccination, follow-up, and blood collections as required by the protocol. A single-arm objective performance criteria for anti-PT antibody seroconversion rates against a 70% target value, where the criterion for success requires the lower limit of the 95% CI for the anti-PT seroconversion rate to exceed 70%. We used the $\chi^2$ test to compare seropositive rates and seroconversion rates of anti-DT and anti-TT between arms. The Wilson (score) method was used to calculate 95% CIs for seropositive rates and seroconversion rates. After $\log_{10}$ transformation of antibody concentrations, GMCs of anti-DT and anti-TT antibodies were compared between arms using the Student's $t$ test. The GMCs and associated 95% CIs were presented. The Wilcox rank-sum test was used to compare differences in median antibody concentrations. The SS included all participants who had received at least one dose of vaccination. The safety outcomes were reported as local and systemic AEs for n (%) and were compared between arms using the $\chi^2$ test or Fisher's exact test. P-value < 0.05 was considered statistically significant. The statistical analysis was performed using SAS 9.4.

## Reporting summary
Further information on research design is available in the Nature Portfolio Reporting Summary linked to this article.

## Data availability
The study protocol is available in the Supplementary Information file. Individual participant data are available under restricted access for the requirements imposed by the Ethics Committee of Zhejiang Provincial Center for Disease Control and Prevention. Researchers who provide a scientifically sound proposal Data and sign a data access agreement will be allowed access to the de-identified individual participant data. Individual participant data can be shared through contacting the corresponding authors. Source data are provided in this paper.

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

## Acknowledgements

We thank the participants in the trial and members of the trial team for their dedication and contributions to the trial. This study was supported by the Technology of Zhejiang Province (2023KY637[XT], 2024KY885[Yao Zhu]) and Public Welfare Technology Application Research Project of Jinhua City (2023-4-248[JC]).

## Author contributions

H.H., Y.T.Z., L.L., J.C., J.Z., and W.Z. contributed to the design of the protocol and design of the study. H.H. is the principal investigator. Y.X., X.T., and J.C. drafted the manuscript. Y.X. and X.T. contributed to the writing of the protocol. X.T., J.C., Y.S., Yang Zhou, R.Y., Yao Zhu, and X.M. contributed to the verification of clinical trial data. H.P., J.Z., W.Z., D.Z., S.L., and L.L. contributed to the investigation and supervision. X.M. contributed to laboratory experimentation. Y.X., X.T., and X.M. accessed and verified the underlying study data. Y.J. did the statistical analysis. All

authors critically reviewed and approved the final version. All authors had full access to the data in the study and had final responsibility for the decision to submit for publication.

## Competing interests

Y.X., L.L., H.P., and Y.T.Z. are employees of the China National Biotec Group Company Limited. J.Z. and D.Z. are employees of the Wuhan Institute of Biological Products Company Limited. S.L. and W.Z. are employees of the Chengdu Institute of Biological Products Company Limited. All other authors declare no competing interests.
