## [Peer Review File · Nature Communications]

Immunogenicity and safety of co-purified diphtheria, tetanus and acellular pertussis vaccine in 6-year-old Chinese children

Corresponding Author: Mr Hanqing He

Version 0:

Reviewer comments:

Reviewer #1

(Remarks to the Author)

The presented study is of significant interest as it addresses the results of a phase IV clinical trial on the booster dose of DTaP (diphtheria, tetanus, and acellular pertussis vaccine) at 6 years of age, in response to the resurgence of pertussis cases in China. The authors provide valuable insights into the immunogenicity and safety of a co-purified DTaP vaccine compared to the DT (diphtheria and tetanus) vaccine in 6-year-old children. However, before the manuscript can be considered for publication, the authors should address the following points to enhance the clarity, depth, and relevance of their work.

Comments:

In the introduction, when discussing the possible causes of the resurgence of pertussis (line 83), the authors should include the waning of immunity induced by current vaccines as a key factor. This is a well-documented phenomenon and is critical to understanding the epidemiology of pertussis in older children and adolescents. Additionally, it would be valuable that the authors explore potential causes of the increased incidence of pertussis in older children following the COVID-19 pandemic. For instance, reduced exposure to *Bordetella pertussis* during lockdowns may have led to diminished natural boosting of immunity, contributing to the observed resurgence.

To facilitate a better interpretation of the results, the authors should provide detailed information on the composition of the pertussis vaccine used in the study. Specifically, they should clarify what is meant by "co-purified" versus "purified" vaccines, the exact quantities of each immunogen (e.g., pertussis toxin [PT], filamentous hemagglutinin [FHA], pertactin [PRN], and fimbriae [FIM]) in the vaccine, and whether the formulation of the booster dose is the same as that used in the primary three-dose series.

Pertussis can present with mild or even asymptomatic symptoms, which may contribute to underdiagnosis and transmission. The authors should clarify how they addressed this aspect in their study. For example, were participants screened for asymptomatic or mild infections prior to vaccination? Could the initial seropositivity divergence among volunteers be attributed to undetected prior exposure to *Bordetella pertussis*?

The study highlights significant variability in baseline seropositivity rates (ranging from 3.57% to 91.45%) and geometric mean concentrations (GMCs) of antibodies against pertussis, diphtheria, and tetanus. The authors should discuss the potential implications of this variability, such as how prior exposure or vaccination history might influence the observed immune responses and what the public health implications are for booster vaccination strategies.

The authors should also acknowledge that pertussis does not have a straightforward correlate of protection. While antibodies play a role in immunity, there are no defined thresholds or specific immune functions that clearly indicate protection. This complexity should be discussed in the context of interpreting seropositivity.

Finally, the authors conclude that a booster dose of DTaP is safe and induces a robust immune response in 6-year-old children, suggesting its feasibility as an alternative to the currently recommended DT vaccine. This recommendation is

supported by the data. However, the authors should also discuss the broader implications of their findings for pertussis control strategies in China and other regions experiencing similar resurgences.

Reviewer #2

(Remarks to the Author)

In their paper Tang et al. report results of a randomised controlled trial comparing the immunogenicity of vaccinating 6-year-old children with either diphtheria-tetanus (DT) or co-purified acellular pertussis DT (DTaP) vaccine. 240 children were randomised to each study arm and serum antibody levels against pertussis (pertussis toxin, PT; filamentous hemagglutinin, FHA), tetanus toxin and diphtheria toxin were measured before and 28 days after vaccination. Study shows that PT and FHA seroconversion rates were higher in children who received DTaP (81.20% and 74.36%) than in children who received DT (2.23% and 3.57%, respectively) without impacting antibody responses against tetanus or diphtheria. Safety profile of DTaP seemed acceptable, even though DTaP induced more mild/moderate adverse events than DT and 1 serious adverse event occurred in DTaP arm of the study.

The paper is easy to understand, the language is scientific and of moderately good quality.

Major comments

1) There is not too much evidence regarding immunogenicity and definitely very little evidence in English regarding the effectiveness of co-purified acellular pertussis vaccine, produced and used in China, against whooping cough. Unfortunately, this study did not exploit the possibility to compare the immunogenicity of co-purified acellular pertussis vaccine with the more widely studied and used antigen-specifically purified acellular pertussis vaccines. Also, it would have been very interesting to see immunogenicity data beyond the first 28 days following the vaccination.

I am highlighting these points to better understand the motivation/scientific question behind this study. Authors state that 6-year booster with co-purified DTaP has been implemented in the national immunisation schedule of China earlier this year. Wouldn't it have been useful for the policy makers to have data in comparison with component purified aPv? My understanding is, that the scarce existing data suggest co-purified vaccine to induce less robust humoral immunity against pertussis than component purified vaccines (Zhiying Yin et al. Comparing the pertussis antibody levels of healthy children immunized with four doses of DTaP-IPV/Hib (Pentaxim) combination vaccine and DTaP vaccine in Quzhou, China. *Frontiers in Immunology* 2023).

Minor comments

- 1) Abstract Line 28-29 "...requiring a booster immunization with the pertussis vaccine." I think this very subjective and depends on what is the aim of the vaccination programme. I would suggest deleting.
- 2) Abstract, lines 51-53. "In response to the resurgence of pertussis in China..." Following the previous comment: I would argue that if the aim of the immunization programme is to protect young infants who are at the biggest risk of pertussis related mortality/morbidity, it would be a priority to study safety and immunogenicity of co-purified DTaP in pregnant women and consider adding this dose to immunization programme. Obviously, if the primary aim of the immunization programme is to reduce disease burden in primary school children, booster dose at 6 years can potentially reduce the disease burden especially 1-2 years following the booster dose.
- 3) Lines 70-72. Again, very different strategies in different countries. Adding booster doses of our current pertussis vaccines, probably cannot completely prevent pertussis epidemics.
- 4) Lines 91-92. As noted in the reference, one reason for increased case numbers in older children, can be increased testing/different testing practices. Any data from China or locally about how much pertussis samples have been taken/positivity rate of samples?
- 5) Lines 109-111. Especially these exclusion criteria: "The exclusion criteria included a history or family history of allergies, convulsions, epilepsy, encephalopathy, or psychosis..." are very broad. Has there been safety concerns with the copurified Tdap regarding these conditions or why was it decided to exclude all these children? Are children with these risk factors still going to be/have been included in the national immunization programme to receive 6-year booster dose?
- 6) Lines 141-142. "After vaccination, the blood sample can be completed within a pre-specified window of 15 days." What does this mean? Does it refer to 1st (pre-vaccination) or 2nd (28 days post-vaccination) blood sample. Anyway, it might skew the results and it would be important to present the data regarding actual sampling timepoints for example in the baseline characteristics table (Table 1).
- 7) Lines 198-199. "480 participants were assessed for eligibility and randomly assigned to either the DTaP or DT groups." I am wondering: There were 0 participants who were assessed and then excluded from the study? That is very unusual especially based on rather extensive exclusion criteria. Any explanations why this might have been?
- 8) Lines 207-208. "... of the participants, including age, gender, ethnic, height, weight, were generally well balanced between the two group ($P>0.05$)." Is there data regarding their primary immunisation? Were all participants vaccinated according to national immunisation schedule? Also, I don't think it is necessary to present P-values for baseline characteristics.
- 9) Lines 238-240. "The incidence of \geq grade 3 vaccine-related adverse events was no significant difference between the two groups (0.42% vs. 0%) (Table 5)." Check wording.
- 10) Lines 242-243. "None of the reported deaths..." Were there deaths in these children during the study? I think they should be mentioned/reported even if they were not accounted as associated with the study intervention.
- 11) Lines 247-248. "With the resurgence of pertussis in China, it seems urgent and necessary to introduce one dose of pertussis vaccine booster immunization for preschool children." I'm referring to my earlier comment. This depends on what is the aim of the immunization programme. Some countries have decided not to implement many boosters for older age groups

because their aim is to prevent deaths in infants, not to reduce coughing illnesses in older children. Their focus is on maternal immunization.

12) Lines 261-263. "As copurified DTaP is only used in a few countries, to our knowledge, our trial is the first to evaluate the immunogenicity and safety of co-purified DTaP in 6-year-old children." I would suggest discussing/citing for example here (or on lines 275-277) a study that compared, though not in an RCT setting, co-purified and antigen-specifically purified aPvs (Zhiying Yin et al. Comparing the pertussis antibody levels of healthy children immunized with four doses of DTap-IPV/Hib (Pentaxim) combination vaccine and DTaP vaccine in Quzhou, China. *Frontiers in Immunology* 2023).

13) Lines 284-289. "According to the data of our trial, we found that after the completion of 4 doses of DTP vaccine, the seropositive rates of antibodies against pertussis and diphtheria have decreased significantly in 6-year-old Chinese children, which decreased to less than 10% against PT and decreased to 30% against FHA and diphtheria. It is suggested that a booster dose of DTaP is needed for children aged 6 years." You don't present any long-term follow-up data. What do you suspect, how long does it take for the PT and FHA antibody levels on average to decay after the booster dose at 6 years? Maybe a follow-up study, outside of this report, at least in a subset of the cohort could be useful?

14) Lines 294-295. I would suggest adding this to results (on lines 244-245) and adding a brief explanation of what Miller-Fisher sdr (belongs to spectrum of Guillain-Barre sdr) is.

15) Tables 2-3. I would suggest using a hyphen rather than a comma in 95% CIs. I would also suggest combining tables 2 and 3 and presenting data also in graph (for example longitudinal boxplots of antibody concentrations for each antigen).

16) Table 4. I find it interesting that adding pertussis component in the vaccine increases reactogenicity of the vaccine to that extent. I have been under impression that tetanus toxoid is the most reactogenic part of the DTP-containing vaccines. Are you aware of any earlier data that you could refer to that would have compared the reactogenicity of (separately purified) DTaP with DT?

Reviewer #3

(Remarks to the Author)

Usually phase 4 trials are observational studies, but this trial is designed like a standard randomized phase 3 trial. It would be very informative if the manuscript explained why this phase 4 trial was designed like a randomized phase 3 trial. The manuscript is poorly written overall, especially in statistics. The study team does not seem to have an experienced statistician. Following are specific comments.

1. "Group" vs. "arm": Treatment groups defined by randomization are called "arms" rather than "groups"
2. The primary endpoint is seroconversion rates. What antibody is this for?
3. LL119-120: This trial used a block randomization with a block size of 8. With a fixed block size, the assignment of every 8th patient will be automatically determined. To avoid this, random block sizes should have been used.
4. LL176-177: "the target value was 70% of the seroconversion rate of anti-pertussis in the DTaP group after vaccination....." => What was the target seroconversion rate of anti-pertussis in the DT arm?
5. L184 and LL190-191: "We used the chi-square test and Fisher's exact test...." => We usually use either chi-square test when N is large or Fisher's exact test when N is small. N=480 is large enough for chi-square test. We do not use the both tests for analyzing one trial. If the both tests were really used in analyzing this trial, we need explanation which test was used for which outcomes and why that test was selected.
6. LL185-186: "the Clopper-Pearson method was used to 186 calculate 95% confidence intervals (CIs)." => For what parameter? Further, Clopper-Pearson CI is for a small N. N=450 for this trial is large enough for a large sample CI method.
7. LL188-189: "...and they are expressed as GMCs with 95% CIs" => This is not a valid statement.
8. LL224-226: What is the p-value for 97.01% vs. 87.05% in seroconversion rate? Also need p-values for seropositive rates of anti-pertussis antibodies and seroconversion rates for anti-PT antibody and anti-FHA antibody.

Version 1:

Reviewer comments:

Reviewer #1

(Remarks to the Author)

The authors have improved the manuscript; however, several critical issues remain unresolved and require further attention.

Please revise the nomenclature of vaccines throughout the manuscript, including in the text, figures, and tables. The Chinese vaccine should be referred to as "co-purified DTaP", while commercially available acellular vaccines may be referred to as "purified commercial DTaP." Additionally, the distinction between purified commercial DTaP and Tdap must be clearly explained, as these formulations differ in both antigen content and target population according to age.

Clarification is needed regarding the composition of the co-purified vaccine. The authors state that pertactin (PRN) is included, but no methodological or compositional data are provided. At a minimum, the manuscript should describe the general principles behind the co-purification process and present supporting evidence for the presence of PRN.

The statistical analysis presented in the figures lacks clarity. Non-significant comparisons (i.e., $p > 0.05$) should be omitted from the figure annotations to avoid confusion and enhance interpretability.

Reviewer comments do not appear to have been clearly integrated into the revised manuscript. Please ensure that all revisions are explicitly highlighted or annotated. In several cases, the authors' responses remain vague or insufficient and must be better reflected and supported in the manuscript text.

Regarding the previously raised comment on the high variability in baseline seropositivity and antibody concentrations (e.g., 3.57% to 91.45%), this issue remains insufficiently addressed. The authors should discuss possible causes of this variability, such as differences in prior exposure, age, vaccination history, or assay sensitivity.

The manuscript references a change in the primary vaccination schedule to 2, 4, and 6 months, yet no rationale or context is provided. Please justify this change, ideally by citing supporting references or official recommendations.

Additionally, the statement in the Discussion that "Our findings provided data on the immunogenicity and safety of co-purified DTaP vaccination in 6-year-old children, which support the adjustment of the DTaP immunization schedule in China" is not clearly supported by the data presented. Please elaborate on how the findings justify this recommendation and specify what kind of adjustment is being proposed (e.g., timing, number of doses, target population).

Reviewer #2

(Remarks to the Author)

Wording here is still not quite clear to me:

"Response: Thank you for your comment. We have revised it in Page 7 Line 158-160 as follows: The incidence of \geq grade 3 vaccine-related adverse events was no significant difference between the two arms (0.83% vs. 0%, $P=0.499$)"

Otherwise my questions/comments have been addressed sufficiently.

Reviewer #3

(Remarks to the Author)

Reviewer #3

4: The authors' response to this comment is inappropriate. When comparing rates between two groups, we need to specify the target rates of both groups for a sample size calculation. That is, we cannot calculate a sample size for a chi-square test (or Fisher exact test) without specifying the expected (or target) rates of both groups.

Version 2:

Reviewer comments:

Reviewer #1

(Remarks to the Author)

While the authors have addressed previous concerns, several pivotal issues require further elaboration to fully solidify the manuscript's arguments and ensure its conclusions are robustly supported.

Major Comments:

- The accurate interpretation of immunogenicity results is contingent upon the baseline health status of participants. The reliance on guardian-reported "no history of pertussis" as a sole inclusion criterion represents a significant methodological limitation, especially given the high community transmission and potential for asymptomatic or mild infections. The authors must clarify:

The specific case definition used to screen and exclude potential participants. Was active surveillance for pertussis-like symptoms conducted in participants or their close contacts prior to enrollment?

Whether the local health system's standardized case definition (incorporating laboratory confirmation, clinical criteria, and/or epidemiological linkage) was applied during the screening process.

The potential for undiagnosed pre-enrollment infection to confound the immunogenicity results—for instance, by inflating the apparent vaccine-induced response in a subset of participants—must be explicitly acknowledged and discussed as a key study limitation.

- The observed lower seroconversion rates among participants who were seropositive at baseline is a critical finding. This topic needs a more profound discussion in the context of current immunological understanding.

For a better interpretation of the results, the authors should explicitly state that, for pertussis, a universally accepted serological correlate of protection remains elusive. While the anti-PT seroconversion rate is encouraging and meets the pre-specified target, the discussion must clearly articulate that the direct translation of these antibody levels into clinical efficacy and durable protection is uncertain.

- The "co-purified" nature of the vaccine is a central differentiator, yet its implications for the final product's composition and contemporary relevance require further clarification. Beyond the stipulation that PT and FHA constitute $\geq 85\%$ of the protein content, the authors should elaborate on the confirmed identity and batch-to-batch consistency of the remaining antigenic components (e.g., PRN, FIM2/3).

- Furthermore, to contextualize the vaccine's current relevance, information on the genetic characteristics of the production

strain (*B. pertussis* Bost strain) and a comparison of its antigenic profile with currently circulating strains in China (e.g., the prevalent ptxP3 genotype) would be highly valuable. A more detailed characterization in the Discussion would facilitate a more meaningful comparison with component-purified vaccines and provide deeper insight into the immune responses observed.

- To further strengthen the safety discussion, the authors should contextualize the reactogenicity findings by providing a direct comparison of adverse event rates (particularly local reactions) with those reported for internationally used Tdap or DTaP booster vaccines in similar age groups. Such a comparison is essential to determine whether the observed reactogenicity profile is unique to this vaccine platform or consistent with acellular pertussis boosters in general.

- The limitations section should be refined for greater precision. The constraint of a single-province study site should be framed not merely as a geographical limitation, but as a potential constraint on the generalizability of the findings to populations with differing background rates of pertussis circulation, natural immunity, and potentially heterogeneous circulating *B. pertussis* strains.

Reviewer #3

(Remarks to the Author)

My concern is appropriately addressed.

REVIEWER COMMENTS

Reviewer #1 (Remarks to the Author):

The presented study is of significant interest as it addresses the results of a phase IV clinical trial on the booster dose of DTaP (diphtheria, tetanus, and acellular pertussis vaccine) at 6 years of age, in response to the resurgence of pertussis cases in China. The authors provide valuable insights into the immunogenicity and safety of a co-purified DTaP vaccine compared to the DT (diphtheria and tetanus) vaccine in 6-year-old children. However, before the manuscript can be considered for publication, the authors should address the following points to enhance the clarity, depth, and relevance of their work.

Comments:

In the introduction, when discussing the possible causes of the resurgence of pertussis (line 83), the authors should include the waning of immunity induced by current vaccines as a key factor. This is a well-documented phenomenon and is critical to understanding the epidemiology of pertussis in older children and adolescents. Additionally, it would be valuable that the authors explore potential causes of the increased incidence of pertussis in older children following the COVID-19 pandemic. For instance, reduced exposure to *Bordetella pertussis* during lockdowns may have led to diminished natural boosting of immunity, contributing to the observed resurgence.

Response: We appreciate it very much for this good suggestion. We have added the causes of the resurgence of pertussis in the Introduction section (Page 4 Line 75-80) as follows:

The reasons for pertussis resurgence in the context of high vaccination coverage are complicated, including the waning of immunity induced by vaccines, the evolution of the pertussis pathogen and advances in detection sensitivity^{12, 13, 14}. In addition, reduced exposure to *Bordetella pertussis* during lockdowns of the COVID-19 pandemic may have led to diminished natural boosting of immunity, contributing to the observed resurgence^{15, 16}.

References

12. Fu P, et al. Pertussis upsurge, age shift and vaccine escape post-COVID-19 caused by ptxP3 macrolide-resistant *Bordetella pertussis* MT28 clone in China. *Clinical microbiology and infection : the official publication of the European Society of Clinical Microbiology and Infectious Diseases* 30, 1439-1446 (2024).
13. Yahong H, Mengyang G, Meng Q, Yu D, Kaihu Y. Rising pertussis cases and deaths in China: current trends and clinical solutions. *Emerging microbes & infections* 13, 2389086 (2024).
14. Liu Y, Yu D, Wang K, Ye Q. Global resurgence of pertussis: A perspective from China. *The Journal of infection* 89, 106289 (2024).
15. Mengyang G, Yahong H, Qinghong M, Wei S, Kaihu Y. Resurgence and atypical patterns of pertussis in China. *The Journal of infection* 88, 106140 (2024).
16. He H, et al. The decline in immunity and circulation of pertussis among Chinese population during the COVID-19 pandemic: A cross-sectional sero-epidemiological study. *Vaccine* 40, 6956-6962 (2022).

To facilitate a better interpretation of the results, the authors should provide detailed information on the composition of the pertussis vaccine used in the study. Specifically, they should clarify what is meant by "co-purified" versus "purified" vaccines, the exact quantities of each immunogen (e.g., pertussis toxin [PT], filamentous hemagglutinin [FHA], pertactin [PRN], and fimbriae [FIM]) in the vaccine, and whether the formulation of the booster dose is the same as that used in the primary three-dose series.

Response: Thank you for your suggestion. We have added the detail information on the composition of the pertussis vaccine used in the trial in the Introduction section (Page 4-5 Line 82-94) as follows: Depending on the manufacturing technique, two types of acellular pertussis vaccines are available globally. One is the more widely used component purified pertussis vaccine, which is mostly used in North America and Europe. Component purified pertussis vaccine has clear contents, usually pertussis toxin (PT) and filamentous hemagglutinin (FHA) as a basis and pertactin (PRN) together with fimbriae (FIM) serotype 2 and serotype 3 making up the remaining ones. The other co-purified pertussis vaccine, which is mostly used in Asian nations such as China and Japan. The components of co-purified pertussis vaccine mainly include PT and FHA, and other *B. pertussis* protective antigens such as PRN, although the exact amount of each antigen in the vaccine is not clear. In China, the most common pertussis vaccine is the diphtheria, tetanus and acellular co-purified pertussis combined vaccine (co-purified DTaP), which has been approved for use in children aged 3 months to 6 years.

We also stated in the Methods section (Page 13 Line 325-327) that the formulation of the booster dose used in this trial is the same as that used in the primary three-dose series as follows: The manufacturing technique and formulation of the co-purified DTaP for 6-year-olds in this trial are the same as China's NIP vaccine used for primary three-dose series at 3, 4, and 5 months old, as well as booster immunization at 18 months.

Pertussis can present with mild or even asymptomatic symptoms, which may contribute to underdiagnosis and transmission. The authors should clarify how they addressed this aspect in their study. For example, were participants screened for asymptomatic or mild infections prior to vaccination? Could the initial seropositivity divergence among volunteers be attributed to undetected prior exposure to *Bordetella pertussis*?

Response: We appreciate it very much for this good suggestion. Given the substantial time and financial costs of screening all participants, we did not screen participants for pertussis infection prior to vaccination. Before enrollment, we asked participants to self-report any history of pertussis, diphtheria, or tetanus. In addition, we also confirmed the disease history of the participants by reviewing the infectious disease report records in the China Information System For Disease Control And Prevention from January 1, 2017 to the day of enrollment.

However, we were unable to screen participants who had a previous or current pertussis infection but did not visit health care facilities for asymptomatic or moderate symptoms. As a limitation of this trial, we added it in the Discussion section (Page 11 Line 265-270) as follow: Second, although guardians were asked to self-report a history of pertussis, diphtheria, or tetanus at enrollment, we did not screen participants for pertussis infection before vaccination. As a result, some participants with asymptomatic or moderate symptomatic pertussis infection who did not visit health care facilities may have been recruited in the trial,

potentially leading to bias in the immunogenicity results.

Since our study was a randomized, controlled trial in which participants with a history of pertussis infection were randomly assigned to either an DTaP group or DT group, the difference in initial seropositivity may not be attributed to undetected prior exposure to Bordetella pertussis.

The study highlights significant variability in baseline seropositivity rates (ranging from 3.57% to 91.45%) and geometric mean concentrations (GMCs) of antibodies against pertussis, diphtheria, and tetanus. The authors should discuss the potential implications of this variability, such as how prior exposure or vaccination history might influence the observed immune responses and what the public health implications are for booster vaccination strategies.

Response: Thank you for your comment.

We have added the seroconversion rates after vaccination for subgroups that were seropositive or negative at baseline (Supplementary Table 1), and the description in the Results section (Page 6 Line 143-148) as follow: In the DTaP, the seroconversion rates after vaccination were higher in participants regardless of whether they were positive or negative for pertussis antibodies at baseline than in the DT, but participants with negative pertussis antibodies at baseline had a higher seroconversion rate than those with positive pertussis antibodies at baseline in the DTaP (Supplemental Table 1).

We have added the potential implications of variability in the Discussion section (Page 9-10 Line 222-236) as follows: According to the data of our trial, it was found that 6-year-olds already had low pertussis antibody seropositive rates before vaccination, with less than 10% against PT and 30% against FHA, and a high seropositive rate of tetanus antibodies before booster vaccination (>90%). This significant variability suggests that a pertussis vaccine booster is needed at age 6 years, the interval between booster vaccination of tetanus vaccine should be considered when using DTaP vaccine as a booster vaccine, or a new monovalent pertussis vaccine should be developed as an alternative. Participants in our trial who received co-purified DTaP had higher seroconversion rates of pertussis antibody than those who received DT, regardless of baseline pertussis antibody levels before booster vaccination. However, participants who were positive for pertussis antibodies at baseline had lower seroconversion rates of PT and FHA after a booster vaccination with co-purified DTaP than those who were negative for pertussis antibodies at baseline, indicating that antibody levels before booster vaccination (e.g., previous exposure or vaccination history) might influence the pertussis immune responses.

The authors should also acknowledge that pertussis does not have a straightforward correlate of protection. While antibodies play a role in immunity, there are no defined thresholds or specific immune functions that clearly indicate protection. This complexity should be discussed in the context of interpreting seropositivity.

Response: Thank you for your reminder. We have added the correlation between pertussis antibodies and protection in the Discussion section (Page 8 Line 187-197) as follows: Limited research has compared the immunogenicity of co-purified DTaP vaccines with that of component-purified DTaP vaccines, the results show that the pertussis antibodies induced by component DTaP vaccines are superior to those of co-purified DTaP vaccine. A study evaluated

the immunogenicity of four doses of DTaP-IPV/Hib and co-purified DTaP in Chinese children and found that the initial GMCs against pertussis after vaccination of DTaP-IPV/Hib was higher than that of co-purified DTaP, declined to similar levels at 72 months after immunization²⁸. However, some studies have also clearly demonstrated that pertussis resurgence has occurred in countries that have administered either co-purified or component-purified vaccines^{29, 30}, confirming that the correlation between pertussis antibody thresholds and protection remains unclear³¹.

References

28. Yin Z, et al. Comparing the pertussis antibody levels of healthy children immunized with four doses of DTaP-IPV/Hib (Pentaxim) combination vaccine and DTaP vaccine in Quzhou, China. *Frontiers in immunology* 13, 1055677 (2022).
29. Hua CZ, He HQ, Shu Q. Resurgence of pertussis: reasons and coping strategies. *World journal of pediatrics : WJP* 20, 639-642 (2024).
30. Campbell PT, McCaw JM, McIntyre P, McVernon J. Defining long-term drivers of pertussis resurgence, and optimal vaccine control strategies. *Vaccine* 33, 5794-5800 (2015).
31. Kapil P, Merkel TJ. Pertussis vaccines and protective immunity. *Current opinion in immunology* 59, 72-78 (2019).

Finally, the authors conclude that a booster dose of DTaP is safe and induces a robust immune response in 6-year-old children, suggesting its feasibility as an alternative to the currently recommended DT vaccine. This recommendation is supported by the data. However, the authors should also discuss the broader implications of their findings for pertussis control strategies in China and other regions experiencing similar resurgences.

Response: Thank you for your suggestion. We have added in the Discussion section (Page 11 Line 282-285) as follows: The data on the use of co-purified DTaP in this study can also provide a reference for countries or regions where pertussis resurgence has occurred, especially those with an increasing proportion of cases among school-aged children, to adjust their immunization strategies.

Reviewer #2 (Remarks to the Author):

In their paper Tang et al. report results of a randomised controlled trial comparing the immunogenicity of vaccinating 6-year-old children with either diphtheria-tetanus (DT) or co-purified acellular pertussis DT (DTaP) vaccine. 240 children were randomised to each study arm and serum antibody levels against pertussis (pertussis toxin, PT; filamentous hemagglutinin, FHA), tetanus toxin and diphtheria toxin were measured before and 28 days after vaccination. Study shows that PT and FHA seroconversion rates were higher in children who received DTaP (81.20% and 74.36%) than in children who received DT (2.23% and 3.57%, respectively) without impacting antibody responses against tetanus or diphtheria. Safety profile of DTaP seemed acceptable, even though DTaP induced more mild/moderate adverse events than DT and 1 serious adverse event occurred in DTaP arm of the study.

The paper is easy to understand, the language is scientific and of moderately good quality.

Major comments

1) There is not too much evidence regarding immunogenicity and definitely very little evidence in English regarding the effectiveness of co-purified acellular pertussis vaccine, produced and used in China, against whooping cough. Unfortunately, this study did not exploit the possibility to compare the immunogenicity of co-purified acellular pertussis vaccine with the more widely studied and used antigen-specifically purified acellular pertussis vaccines. Also, it would have been very interesting to see immunogenicity data beyond the first 28 days following the vaccination.

Response: Thank you for your comment. In China, only one component purified pertussis vaccine (Sanofi Pasteur's DTap-IPV/Hib combination vaccine) has been approved and is used for infants and toddlers in a four-dose schedule, with primary immunization at 2, 3, 4 months of age or 3, 4, 5 months of age, followed by a booster dose at 18 months. The tetanus toxoid, reduced diphtheria toxoid, and acellular pertussis (Tdap) vaccine, which is commonly used in adolescents, adults, and pregnant women in North America and Europe, has also not been approved for use in China. As a result, it is unfortunate that we were unable to compare the immunogenicity of the co-purified pertussis vaccine to that of the component purified pertussis vaccine following booster immunization in 6-year-old children because no component purified pertussis vaccine approved for use in 6-year-old children is available. We have revised the texts in the Introduction section (Page 4 Line 82-105) as follows: Depending on the manufacturing technique, two types of acellular pertussis vaccines are available globally. One is the more widely used component purified pertussis vaccine, which is mostly used in North America and Europe. Component purified pertussis vaccine has clear contents, usually pertussis toxin (PT) and filamentous hemagglutinin (FHA) as a basis and pertactin (PRN) together with fimbriae (FIM) serotype 2 and serotype 3 making up the remaining ones. The other co-purified pertussis vaccine, which is mostly used in Asian nations such as China and Japan. The components of co-purified pertussis vaccine mainly include PT and FHA, and other B. pertussis protective antigens such as PRN, although the exact amount of each antigen in the vaccine is not clear. In China, the most common pertussis vaccine is the diphtheria, tetanus and acellular co-purified pertussis combined vaccine (co-purified DTaP), which has been approved for use in children aged 3 months to 6 years. Before January 1, 2025, the China's National Immunization Program (NIP) recommending a four-dose of co-purified DTaP schedule at 3, 4, 5, and 18 months. In addition to co-purified DTaP, there are two non-NIP DTaP vaccines (i.e., private, out-of-pocket expense) available for infants and toddlers in China: a pentavalent DTaP-IPV/Hib combination vaccine comprising component purified pertussis vaccine, and a quadrivalent DTaP-Hib combination vaccine comprising co-purified pertussis vaccine. In certain countries such as the United States, the United Kingdom, and Canada, after completing four doses of pertussis vaccine for children under the age of two, booster doses of pertussis vaccine are recommended for children 4-6 years old, adolescents, pregnant women, and adults^{19, 20, 21}. However, a new component pertussis vaccine has not been licensed for use in China in children 6 years of age or older, adolescents, adults, or pregnant women.

References

19. Gidengil CA, Sandora TJ, Lee GM. Tetanus-diphtheria-acellular pertussis vaccination of adults in the USA. Expert review of vaccines 7, 621-634 (2008).
20. Amirthalingam G, et al. Optimization of Timing of Maternal Pertussis Immunization From

6 Years of Postimplementation Surveillance Data in England. *Clinical infectious diseases* : an official publication of the Infectious Diseases Society of America 76, e1129-e1139 (2023).

21. MacDougall D, et al. Universal tetanus, diphtheria, acellular pertussis (Tdap) vaccination of adults: What Canadian health care providers know and need to know. *Human vaccines & immunotherapeutics* 11, 2167-2179 (2015).

I am highlighting these points to better understand the motivation/scientific question behind this study. Authors state that 6-year booster with co-purified DTaP has been implemented in the national immunisation schedule of China earlier this year. Wouldn't it have been useful for the policy makers to have data in comparison with component purified aPv? My understanding is, that the scarce existing data suggest co-purified vaccine to induce less robust humoral immunity against pertussis than component purified vaccines (Zhiying Yin et al. Comparing the pertussis antibody levels of healthy children immunized with four doses of DTaP-IPV/Hib (Pentaxim) combination vaccine and DTaP vaccine in Quzhou, China. *Frontiers in Immunology* 2023).

Response: Thank you for your comment and suggestion. Just as you said, the limited studies that have been conducted suggest that the immunogenicity of components-purified pertussis vaccinations is superior to that of co-purified vaccines. However, in China, there is no component-purified DTaP vaccine available for 6-year-olds. Therefore, we were unable to conduct a comparative study between the co-purified DTaP vaccine and the component-purified DTaP vaccine. At the time of our trial (2023), China had not implemented the booster immunization strategy for 4-6-year-olds that many other countries had used. This trial was based on the co-purified DTaP vaccination, which is the only one available for 6-year-old children in China. Chinese vaccine manufacturers are accelerating the research progress of component purified pertussis vaccines and combination vaccines, so that in the future, component purified pertussis vaccines may be used instead of co-purified vaccinations in China's NIP. We have added the texts in the Discussion section (Page 8 Line 184-193) as follows: Since co-purified DTaP is only used in a few countries, and there is no component-purified DTaP vaccine available for 6-year-olds in China, our trial was the first to evaluate the immunogenicity and safety of co-purified DTaP in 6-year-old children. Limited research has compared the immunogenicity of co-purified DTaP vaccines with that of component-purified DTaP vaccines, the results show that the pertussis antibodies induced by component DTaP vaccines are superior to those of co-purified DTaP vaccine. A study evaluated the immunogenicity of four doses of DTaP-IPV/Hib and co-purified DTaP in Chinese children and found that the initial GMCs against pertussis after vaccination of DTaP-IPV/Hib was higher than that of co-purified DTaP, declined to similar levels at 72 months after immunization²⁸.

References

28. Yin Z, et al. Comparing the pertussis antibody levels of healthy children immunized with four doses of DTaP-IPV/Hib (Pentaxim) combination vaccine and DTaP vaccine in Quzhou, China. *Frontiers in immunology* 13, 1055677 (2022).

Minor comments

1) Abstract Line 28-29 "...requiring a booster immunization with the pertussis vaccine." I think this very subjective and depends on what is the aim of the vaccination programme. I would suggest

deleting.

Response: Revised as the reviewer's suggestion.

2) Abstract, lines 51-53. "In response to the resurgence of pertussis in China..." Following the previous comment: I would argue that if the aim of the immunization programme is to protect young infants who are at the biggest risk of pertussis related mortality/morbidity, it would be a priority to study safety and immunogenicity of co-purified DTaP in pregnant women and consider adding this dose to immunization programme. Obviously, if the primary aim of the immunization programme is to reduce disease burden in primary school children, booster dose at 6 years can potentially reduce the disease burden especially 1-2 years following the booster dose.

Response: We agree with your point. In China, in addition to adding one dose for 6-year-olds, the pertussis vaccine schedule has also been adjusted from 3, 4, and 5 months to 2, 4, and 6 months, extending protection period for babies at younger ages. This adjustment was supported by the results of another clinical study and is not reported in the manuscript. In addition, as mentioned in the previous response, no pertussis vaccine has been approved for use in pregnant women or adolescents or adults in China, which limits policy decisions. The booster dose of the DTaP vaccine for 6-year-old children is mainly to reduce the disease burden among school-aged children. We have revised the texts in the Abstract section (Page 2 Line 49-52) as follows: In response to the high incidence of pertussis among school-aged children over 6 years old in China, it may be feasible to give a booster dose of co-purified DTaP instead of the recommended DT for 6-year-old children.

3) Lines 70-72. Again, very different strategies in different countries. Adding booster doses of our current pertussis vaccines, probably cannot completely prevent pertussis epidemics.

Response: Thank you for your comment. "Since neither natural infection nor pertussis vaccination may confer lifetime protection, regular booster immunization with pertussis vaccination is a sound strategy." This sentence's expression is uncertain, therefore we delete it based on context.

4) Lines 91-92. As noted in the reference, one reason for increased case numbers in older children, can be increased testing/different testing practices. Any data from China or locally about how much pertussis samples have been taken/positivity rate of samples?

Response: Thank you for your comment. In China, PCR have been incorporated into the national diagnostic criteria for pertussis by the end of 2023, greatly shortening the detection time and improving the detection rate, which may be one of the reasons for the increased case numbers in older children. Through literature review, we found that there were few data on pertussis samples have been taken/positivity rate of samples from China. Between 2017 and 2018, a sero-epidemiological survey was conducted in five Chinese provinces, recruiting 6060 healthy subjects. It was found that sero-estimated recent infection rates appear to grow from school age into adolescence and adulthood (Zhang Z, Wang Q, Zhu Q, et al. Seroepidemiology of pertussis immunity in five provinces of China: A population-based, cross-sectional study. Hum Vaccin Immunother. 2024 Dec 31;20(1):2417532). Another analysis of pertussis case data showed that the incidence of pertussis in the age group of 5-9 years old in Chaoyang District of Beijing was the highest

(24.13/100,000) in 2023-2024(Liu F, Li Z, Hu Y, et al. The recent rapid rise in pertussis in Chaoyang District, Beijing: Improved recognition and diagnostic capabilities. *J Infect.* 2024 Nov;89(5):106272.). **We have added references in the Introduction (Page 5 Line 106-109) as follows:** Recent epidemiological studies of pertussis in China have revealed a significant rise in the frequency of pertussis cases in children over the age of 6 years^{22,23,24,25,26}, indicating that preschool children should be immunized with one dose of pertussis vaccine.

References

22. Liu Y, Ye Q. Resurgence and the shift in the age of peak onset of pertussis in southern China. *The Journal of infection* 89, 106194 (2024).

23. Liu F, Li Z, Hu Y, Jia B, Ma J. The recent rapid rise in pertussis in Chaoyang District, Beijing: Improved recognition and diagnostic capabilities. *The Journal of infection* 89, 106272 (2024).

24. Zhang Z, et al. Seroepidemiology of pertussis immunity in five provinces of China: A population-based, cross-sectional study. *Human vaccines & immunotherapeutics* 20, 2417532 (2024).

25. Dai H, et al. Underestimated Incidence Rate of Pertussis in the Community: Results from Active Population-Based Surveillance in Yiwu, China. *Microorganisms* 12, (2024).

26. Zhu Y, et al. Seroprevalence of IgG antibodies against pertussis toxin in the Chinese population: A systematic review and meta-analysis. *Human vaccines & immunotherapeutics* 20, 2341454 (2024).

5) Lines 109-111. Especially these exclusion criteria: “The exclusion criteria included a history or family history of allergies, convulsions, epilepsy, encephalopathy, or psychosis...” are very broad. Has there been safety concerns with the copurified Tdap regarding these conditions or why was it decided to exclude all these children? Are children with these risk factors still going to be/have been included in the national immunization programme to receive 6-year booster dose?

Response: Thank you for your comment. Our trial was conducted in 2023, when children aged 6 years in China were vaccinated with DT vaccine, and there was no previous experience in the use of co- purified DTaP vaccine in children aged 6 years, especially in terms of vaccine safety. Out of prudence, we developed the "exclusion criteria" based on the vaccination package insert and used broad exclusion criteria to ensure that the participants were in good health. The coverage rate of Chinese NIP vaccine is required to be more than 90%, so most children aged 6 years can be vaccinated with co- purified DTaP vaccine. For children with these risk factors, the clinicians at vaccination clinics need to refer to the package insert of the vaccine and weigh the benefits of vaccination to make a decision on vaccination. At present, there is no real-world data disclosure of DTaP vaccine for children aged 6 years in China. We will continue to pay attention to the safety and immunogenicity of co-purified DTaP vaccination for children with special health status. We have added the description in the Discussion (Page 9 Line 208-212) as follows: The aim of this study was to evaluate the immunogenicity and safety of co-purified DTaP vaccination in health children at age 6 years, but and the vaccine effectiveness, immunogenicity and safety of a booster dose at age 6 years needs to be further explored in larger real-world populations (including special health populations) in China.

6) Lines 141-142. “After vaccination, the blood sample can be completed within a pre-specified window of 15 days.” What does this mean? Does it refer to 1st (pre-vaccination) or 2nd (28 days post-vaccination) blood sample. Anyway, it might skew the results and it would be important to present the data regarding actual sampling timepoints for example in the baseline characteristics table (Table 1).

Response: We apologize for the confusion. We have revised it in Page 13 Line 335-336 as follows: Blood sample collection after vaccination can be completed within 28-42 days.

7) Lines 198-199. “480 participants were assessed for eligibility and randomly assigned to either the DTaP or DT groups.” I am wondering: There were 0 participants who were assessed and then excluded from the study? That is very unusual especially based on rather extensive exclusion criteria. Any explanations why this might have been?

Response: I apologize for confusing you because our description is not clear. Participants in this trial were required to have completed four doses of DTaP vaccine, which means that these children had no contraindications to receiving these four doses. Prior to recruitment, 6-year-olds in the local community were prescreened again by the study staff for vaccine contraindications in the vaccination information system. In fact, a total of 527 participants received informed consent, physical examination, and exclusion criteria, and 480 of them were enrolled in the study. We have revised it in Page 5 Line 114-115 as follows: 527 participants were assessed for eligibility and 480 participants randomly assigned to either the DTaP or DT arms. **We also revised the Figure 1.**

8) Lines 207-208. “... of the participants, including age, gender, ethnic, height, weight, were generally well balanced between the two group ($P>0.05$).” Is there data regarding their primary immunisation? Were all participants vaccinated according to national immunisation schedule? Also, I don't think it is necessary to present P-values for baseline characteristics.

Response: Thank you for your comment. We do not have the data about primary immunization. Not all participants were vaccinated according to the Chinese's NIP schedule. In addition to co-purified DTaP, there are two non-EPI DTaP vaccines (i.e., private, out-of-pocket expense) available in China: a pentavalent DTaP-IPV/Hib combination vaccine comprising component purified pertussis vaccine, and a quadrivalent DTaP-Hib combination vaccine comprising co-purified pertussis vaccine. To be more consistent with real-world usage settings, we did not exclude children who had previously received pentavalent or quadrivalent vaccines. Thus, it is possible that participants who received the pentavalent vaccine may have received it at 2, 3, and 4 months of age rather than at 3, 4, and 5 months of age as in the NIP schedule.

We have added the following explanation in the Introduction section (Page 5 Line 96-100) as follows: In addition to co-purified DTaP, there are two non-NIP DTaP vaccines (i.e., private, out-of-pocket expense) available for infants and toddlers in China: a pentavalent DTaP-IPV/Hib combination vaccine comprising component purified pertussis vaccine, and a quadrivalent DTaP-Hib combination vaccine comprising co-purified pertussis vaccine.

9) Lines 238-240. “The incidence of \geq grade 3 vaccine-related adverse events was no significant difference between the two groups (0.42% vs. 0%) (Table 5).” Check wording.

Response: Thank you for your comment. We have revised it in Page 7 Line 158-160 as follows: The incidence of \geq grade 3 vaccine-related adverse events was no significant difference between the two arms (0.83% vs. 0%, $P=0.499$) (Table 4).

10) Lines 242-243. “None of the reported deaths...” Were there deaths in these children during the study? I think they should be mentioned/reported even if they were not accounted as associated with the study intervention.

Response: We apologize for the confusion. We have revised it in Page 7 Line 162-163 as follows: There were no deaths among any participants during the study period.

11) Lines 247-248. “With the resurgence of pertussis in China, it seems urgent and necessary to introduce one dose of pertussis vaccine booster immunization for preschool children.” I’m referring to my earlier comment. This depends on what is the aim of the immunization programme. Some countries have decided not to implement many boosters for older age groups because their aim is to prevent deaths in infants, not to reduce coughing illnesses in older children. Their focus is on maternal immunization.

Response: Thank you for your comment. As we answered in the previous comment, we agree with your point. In China, no pertussis vaccine has been approved for use in children 6 years of age or older, adolescents, adults, or pregnant women in China, which limits policy decisions.

12) Lines 261-263. “As copurified DTaP is only used in a few countries, to our knowledge, our trial is the first to evaluate the immunogenicity and safety of co-purified DTaP in 6-year-old children.” I would suggest discussing/citing for example here (or on lines 275-277) a study that compared, though not in an RCT setting, co-purified and antigen-specifically purified aPvs (Zhiying Yin et al. Comparing the pertussis antibody levels of healthy children immunized with four doses of DTaP-IPV/Hib (Pentaxim) combination vaccine and DTaP vaccine in Quzhou, China. *Frontiers in Immunology* 2023).

Response: Thank you for your suggestion. We have added in Page 8 Line 190-193 as follows: A study evaluated the immunogenicity of four doses of DTaP-IPV/Hib and co-purified DTaP in Chinese children and found that the initial GMCs against pertussis after vaccination of DTaP-IPV/Hib was higher than that of co-purified DTaP, declined to similar levels at 72 months after immunization²⁸.

References

28. Yin Z, et al. Comparing the pertussis antibody levels of healthy children immunized with four doses of DTaP-IPV/Hib (Pentaxim) combination vaccine and DTaP vaccine in Quzhou, China. *Frontiers in immunology* 13, 1055677 (2022).

13) Lines 284-289. “According to the data of our trial, we found that after the completion of 4 doses of DTP vaccine, the seropositive rates of antibodies against pertussis and diphtheria have decreased significantly in 6-year-old Chinese children, which decreased to less than 10% against PT and decreased to 30% against FHA and diphtheria. It is suggested that a booster dose of DTaP is needed for children aged 6 years.” You don’t present any long-term follow-up data. What do you suspect, how long does it take for the PT and FHA antibody levels on average to decay after the booster dose at 6 years? Maybe a follow-up study, outside of this report, at least in a subset of the cohort could

be useful?

Response: We apologize for the confusion. We would like to express that the results of this trial found that 6-year-olds already had low pertussis antibody seropositive rates before vaccination.

We have revised it in Page 9 Line 222-229 as follows: According to the data of our trial, it was found that 6-year-olds already had low pertussis antibody seropositive rates before vaccination, with less than 10% against PT and 30% against FHA, and a high seropositive rate of tetanus antibodies before booster vaccination (>90%). This significant variability suggests that a pertussis vaccine booster is needed at age 6 years, the interval between booster vaccination of tetanus vaccine should be considered when using DTaP vaccine as a booster vaccine, or a new monovalent pertussis vaccine should be developed as an alternative.

14) Lines 294-295. I would suggest adding this to results (on lines 244-245) and adding a brief explanation of what Miller-Fisher sdr (belongs to spectrum of Guillain-Barre sdr) is.

Response: Thank you for your suggestion. We have revised as the reviewer's suggestion in Page 7 Line 164-168: A 6-year-old boy developed fever three days after co-purified DTaP vaccination and was hospitalized for a diagnosis of Miller-Fisher syndrome (MFS) that as a variant of Guillain-Barré syndrome, due to limb weakness and binocular strabismus. This SAE was reported in the DTaP, which was considered unrelated to the vaccine.

15) Tables 2-3. I would suggest using a hyphen rather than a comma in 95% CIs. I would also suggest combining tables 2 and 3 and presenting data also in graph (for example longitudinal boxplots of antibody concentrations for each antigen).

Response: Revised as the reviewer's suggestion. We have combined the Table 2 and Table 3, and added a figure for each antigen antibody concentrations (Figure 2).

16) Table 4. I find it interesting that adding pertussis component in the vaccine increases reactogenicity of the vaccine to that extent. I have been under impression that tetanus toxoid is the most reactogenic part of the DTP-containing vaccines. Are you aware of any earlier data that you could refer to that would have compared the reactogenicity of (separately purified) DTaP with DT?

Response: We appreciate it very much for this good suggestion. In 2009-2010, GSK's Tdap (Boostrix) was briefly licensed for use in children aged 6-8 years in China. One study conducted in China showed comparable safety data between GSK's Tdap and DT in 6-year-old children (Zhu F, Zhang S, Hou Q, et al. Booster vaccination against pertussis in Chinese children at six years of age using reduced antigen content diphtheria-tetanus-acellular pertussis vaccine (Boostrix). Hum Vaccin. 2010 Mar 3;6(3):10503.). We have added it in Page 10 Line 240-243: A study conducted in China in 2007 showed that the safety data of the tetanus-diphtheria-acellular pertussis (Tdap) vaccine, which was briefly approved in China for children aged 6 to 8, was comparable to that of the DT vaccine among 6-year-old children³⁶.

References

36. Zhu F, et al. Booster vaccination against pertussis in Chinese children at six years of age using reduced antigen content diphtheria-tetanus-acellular pertussis vaccine (Boostrix()). Human vaccines 6, (2010).

In addition, according to the Adverse event following immunization (AEFI) passive

surveillance data released by the Chinese Center for Disease Control and Prevention, the incidences of reported AEFIs were 80.58 cases per 100 000 doses for co-purified DTaP, 49.71 cases per 100 000 doses for DT, and 5.62 cases per 100 000 doses for tetanus vaccine in 2021-2022. The incidences of reported AEFIs for co-purified DTaP was higher than that of DT after large-scale use in China, which was consistent with our trial's safety findings. We have added it in Page 10 Line 243-248: According to the Adverse event following immunization (AEFI) passive surveillance data released by the Chinese Center for Disease Control and Prevention during 2021-2022, the incidences of reported AEFIs were 80.58 cases per 100 000 doses for co-purified DTaP, 49.71 cases per 100 000 doses for DT³⁷. The incidences of reported AEFIs for co-purified DTaP was higher than that of DT after large-scale use in China, which was consistent with our safety findings.

References

37. Zhang L, et al. Surveillance of adverse events following immunization in China, 2021-2022. Chinese Journal of Vaccines and Immunization 30, 470-484 (2024).

Reviewer #3 (Remarks to the Author):

Usually phase 4 trials are observational studies, but this trial is designed like a standard randomized phase 3 trial. It would be very informative if the manuscript explained why this phase 4 trial was designed like a randomized phase 3 trial. The manuscript is poorly written overall, especially in statistics. The study team does not seem to have an experienced statistician. Following are specific comments.

Response: Thank you very much for your suggestions and comments. The aim of this study was to compare the immunogenicity and safety of co-purified DTaP vaccine with the DT vaccine recommended by the Chinese's NIP in 6-year-old children. To control the potential confounding factors that may affect the results, we adopted a randomized controlled clinical trial method and randomly assigned the children to two arms to receive the two types of vaccines respectively. We have added relevant explanations in the Methods section (Page 11-12 Line 288-294) as follows: To compare the immunogenicity and safety of the co-purified DTaP with the DT recommended by China's NIP in 6-year-old children, a randomized, controlled, open-label, phase 4 trial was conducted to control potential confounding factors that might affect the results. From April to July 2023, our trial was conducted in the Fuyang District of Hangzhou City and Tongxiang City of Zhejiang Province, China. This trial was approved by the Ethics Committee of Zhejiang Provincial Center for Disease Control and Prevention (2023-008-1).

We have invited experienced statisticians to consult on the revision of the statistical analysis in the manuscript, and we have provided a point-by-point response to your comments.

1. "Group" vs. "arm": Treatment groups defined by randomization are called "arms" rather than "groups"

Response: Thank you for your suggestion. We have revised "group" to "arm" in the manuscript.

2. The primary endpoint is seroconversion rates. What antibody is this for?

Response: Thank you for your comment. The co-purified DTaP vaccine induces antibody responses against pertussis (PT and FHA), diphtheria and tetanus. The DT vaccine induces antibody responses against diphtheria and tetanus. Therefore, the primary endpoint in our study were seroconversion rates of antibodies against pertussis, diphtheria, and tetanus with DTaP vaccine (in DTaP arm) and against diphtheria and tetanus with DT vaccine (in DT arm).

3. LL119-120: This trial used a block randomization with a block size of 8. With a fixed block size, the assignment of every 8th patient will be automatically determined. To avoid this, random block sizes should have been used.

Response: We appreciate it very much for this good suggestion. We strongly agree with your suggestion that the randomized block size could reduce predictability in our unblinded trial. We have added it as a limitation in the Discussion section (Page 11 Line 270-273) as follows: Finally, our trial used a block randomization with a block size of 8 rather than random block size, which increased the predictability. Nevertheless, the results showed that the baseline characteristics of the participants were comparable between the two arms. Thank you for reminding us again, random block sizes will be considered in accordance with the design in our future clinical studies.

4. LL176-177: “the target value was 70% of the seroconversion rate of anti-pertussis in the DTaP group after vaccination.....” => What was the target seroconversion rate of anti-pertussis in the DT arm?

Response: Thank you for your comment. Because the DT vaccine does not contain pertussis antigen and theoretically does not induce antibody responses against pertussis, therefore, we did not set a target seroconversion rate of anti-pertussis in the DT arm.

5. L184 and LL190-191: “We used the chi-square test and Fisher’s exact test....” => We usually use either chi-square test when N is large or Fisher’s exact test when N is small. N=480 is large enough for chi-square test. We do not use the both tests for analyzing one trial. If the both tests were really used in analyzing this trial, we need explanation which test was used for which outcomes and why that test was selected.

Response: Thank you very much for pointing out this error and apologize for that. We only performed the chi-square test as you described. We have revised it in Page 14 Line 377-378 as follows: We used the χ^2 test to compare seropositive rates and seroconversion rates between arms.

6. LL185-186: “he Clopper-Pearson method was used to 186 calculate 95% confidence intervals (CIs).” => For what parameter? Further, Clopper-Pearson CI is for a small N. N=450 for this trial is large enough for a large sample CI method.

Response: Thank you for your suggestion. Since Clopper-Pearson method might be too conservative for a big sample size, we have used Wilson (score) method to re-calculate the 95% CIs for seropositive rates and seroconversion rates (Table 2). And we have revised the description in Page 14 Line 378-379 as follows: The Wilson (score) method was used to

calculate 95% confidence intervals (CIs) for seropositive rates and seroconversion rates.

7. LL188-189: "...and they are expressed as GMCs with 95% CIs" => This is not a valid statement.

Response: Thank you very much for pointing out this error and apologize for that. We have revised the description in Page 15 Line 381 as follows: The GMC and associated 95% CIs were presented.

8. LL224-226: What is the p-value for 97.01% vs. 87.05% in seroconversion rate? Also need p-values for seropositive rates of anti-pertussis antibodies and seroconversion rates for anti-PT antibody and anti-FHA antibody.

Response: Thank you for your suggestion. We have added the P values in the Results section.

Reviewer #1 (Remarks to the Author):

The authors have improved the manuscript; however, several critical issues remain unresolved and require further attention.

Response: Thanks for the positive comments. We hope that our point-to-point revision would satisfactorily address the comments and concerns of the reviewer.

Please revise the nomenclature of vaccines throughout the manuscript, including in the text, figures, and tables. The Chinese vaccine should be referred to as “co-purified DTaP”, while commercially available acellular vaccines may be referred to as “purified commercial DTaP.” Additionally, the distinction between purified commercial DTaP and Tdap must be clearly explained, as these formulations differ in both antigen content and target population according to age.

Response: Thank you for your suggestions on the distinctions in the nomenclature of different pertussis vaccines in the manuscript. Indeed, the co-purified DTaP vaccine is used in China's NIP, while the majority of countries use the component purified DTaP vaccine (component DTaP). Following expert advice, we have uniformly modified "co-purified DTaP" and "component DTaP" throughout the manuscript text, figures, and tables for differentiation.

We have added explanations of co-purified DTaP, component DTaP and Tdap in the Introduction section (Page 4-5 Line 91-96) as follows: Based on differences in target population and antigen content, component-purified pertussis vaccines that are more commonly used are the component-purified diphtheria, tetanus and acellular pertussis vaccine (component DTaP) for infants and young children, and the reduced antigen content diphtheria, tetanus, and acellular pertussis vaccine (Tdap) mainly for adolescents, adults, and pregnant women.

And we have revised the texts in the Introduction section (Page 5 Line 112-120) as follows:

In certain countries, such as the United States, Canada and South Korea, after completing four doses of pertussis vaccine for children under the age of two, it is recommended that children aged 4 to 6 receive a booster dose of component DTaP or Tdap, and that adolescents, pregnant women, and adults also receive a booster dose of Tdap^{21, 22, 23, 24}. However, no new component pertussis vaccine (component DTaP or Tdap) has been licensed for use in China in children aged 6 and above, adolescents, adults, or pregnant women.

Reference

21. Havers FP, Moro PL, Hunter P, Hariri S, Bernstein H. Use of Tetanus Toxoid, Reduced Diphtheria Toxoid, and Acellular Pertussis Vaccines: Updated Recommendations of the Advisory Committee on Immunization Practices - United States, 2019. MMWR Morbidity and mortality weekly report 69, 77-83 (2020).
22. Gidengil CA, Sandora TJ, Lee GM. Tetanus-diphtheria-acellular pertussis vaccination of adults in the USA. Expert review of vaccines 7, 621-634 (2008).
23. MacDougall D, et al. Universal tetanus, diphtheria, acellular pertussis (Tdap) vaccination of adults: What Canadian health care providers know and need to know. Human vaccines & immunotherapeutics 11, 2167-2179 (2015).

24. Choi KM, et al. Recommendation for the use of newly introduced Tdap vaccine in Korea. Korean journal of pediatrics 54, 141-145 (2011).

Clarification is needed regarding the composition of the co-purified vaccine. The authors state that pertactin (PRN) is included, but no methodological or compositional data are provided. At a minimum, the manuscript should describe the general principles behind the co-purification process and present supporting evidence for the presence of PRN.

Response: Thank you for your constructive suggestion. We agree that clarification regarding the composition of the co-purified DTaP vaccine is essential.

In accordance with Chinese regulatory requirements, a mandatory lot release system has been implemented for all vaccines, ensuring that no vaccine batch enters the market without passing rigorous quality control (Reference: Wang L, Lei D, Zhang S. Acellular pertussis vaccines in China. *Vaccine*. 2012;30:7174–7178). As the designated national authority for vaccine lot release, the National Institutes for Food and Drug Control (NIFDC) also serves as the antibody testing institution for our study. Following NIFDC's standard protocols, only anti-PT and anti-FHA antibody data from the co-purified DTaP are routinely reported. Consequently, our study exclusively presents the antibody responses for these two key antigens.

In this literature²⁰, the National Institutes of Food and Drug Control (NIFDC) analyzed the composition of the co-purified DTaP based on 2-dimensional gel electrophoresis-based proteomic technology. The results showed that in addition to the main antigens PT and FHA, the co-purified DTaP also contained PRN and other minor protein antigens.

The pertussis antigen components of the co-purified DTaP are mainly PT and FHA (accounting for at least 85%), and also contain other antigen components. The specific content of each component is unknown.

In response, we have added references in the introduction section (Page 5 Line 97-99) regarding the co-purified DTaP vaccine containing PRN components as follows: The components of co-purified pertussis vaccine mainly include PT and FHA, and other B. pertussis protective antigens such as PRN, although the exact amount of each antigen in the vaccine is not clear²⁰.

Reference

20. Xu Y, Tan Y, Asokanathan C, Zhang S, Xing D, Wang J. Characterization of co-purified acellular pertussis vaccines. *Human vaccines & immunotherapeutics* 11, 421-427 (2015).

We also have added the relevant content about the co-purification process of pertussis vaccine in the Methods section (Page 14 Line 368-377) as follows: The general principles of the co-purification process: The Bordetella pertussis Bost strain was cultured via fermentation to generate bacterial harvest containing key pertussis antigenic components: PT, FHA, PRN, and FIM2/3, etc. Following co-purification via ammonium sulfate precipitation, extraction, and sucrose density gradient centrifugation, the pertussis bacterial harvest was purified to yield a refined antigen solution containing multiple target immunogens. Chemical

detoxification with glutaraldehyde generated the pertussis bulk solution, which was used to prepare the co-purified DTaP. The Chinese Pharmacopoeia stipulates that for co-purified DTaP vaccines, the content of PT and FHA shall constitute no less than 85% of the total protein content.

The statistical analysis presented in the figures lacks clarity. Non-significant comparisons (i.e., $p > 0.05$) should be omitted from the figure annotations to avoid confusion and enhance interpretability.

Response: Thank you for your valuable suggestion. We have revised Figure 2 in accordance with your suggestions.

Previous Figure 2:

Modified Figure 2:

Reviewer comments do not appear to have been clearly integrated into the revised manuscript. Please ensure that all revisions are explicitly highlighted or annotated. In several cases, the authors' responses remain vague or insufficient and must be better reflected and supported in the manuscript text.

Response: We sincerely apologize for not clearly indicating the revision status in the previous version. Thank you for this important feedback. In this revised version, all modifications in the manuscript text have been highlighted in blue.

Regarding the previously raised comment on the high variability in baseline seropositivity and antibody concentrations (e.g., 3.57% to 91.45%), this issue remains insufficiently addressed. The authors should discuss possible causes of this variability, such as differences in prior exposure, age, vaccination history, or assay sensitivity.

Response: Thank you very much for your thoughtful suggestions. Indeed, our study found the high variability in baseline seropositivity and antibody concentrations (e.g., 3.57% to 91.45%). A study of 6- to 8-year-old children in China who had completed 4 doses of co-purified DTwP showed the following pre-booster seropositivity rates: 5.6% for anti-PT, 40.1% for anti-FHA, 72.2% for anti-DT, and 83.3% for anti-TT, also demonstrating high variability in seropositivity rates between different antigens (5%–83%). All children enrolled in this trial were needed to have received four doses of the co-purified DTaP vaccination and self-reported no history of illness with related diseases. As a result, the reason for the high variation of seropositive rates for three antibodies among 6-year-old children might be attributed to different antigen characteristics and detection sensitivity of vaccines.

We have added the relevant content in the Discussion section (Page 10 Line 250-259) as

follows: A study of 6- to 8-year-old children in China who had completed 4 doses of co-purified DTwP showed the following pre-booster seropositivity rates: 5.6% for anti-PT, 40.1% for anti-FHA, 72.2% for anti-DT, and 83.3% for anti-TT, also demonstrating high variability in seropositivity rates between different antigens (5%–83%)³⁹. All children enrolled in this trial were needed to have received four doses of the co-purified DTaP vaccination and self-reported no history of illness with related diseases. As a result, the reason for the high variation of seropositive rates for three antibodies among 6-year-old children might be attributed to different antigen characteristics and detection sensitivity of vaccines.

Reference

39. Zhu F, et al. Booster vaccination against pertussis in Chinese children at six years of age using reduced antigen content diphtheria-tetanus-acellular pertussis vaccine (Boostrix()). *Human vaccines* 6, (2010).

The manuscript references a change in the primary vaccination schedule to 2, 4, and 6 months, yet no rationale or context is provided. Please justify this change, ideally by citing supporting references or official recommendations.

Response: We acknowledge the feedback and regret the insufficient contextualization of the background and rationale for China's co-purified DTaP immunization schedule adjustment.

The adjustment of the immunization schedule for the co-purified DTaP in China by 2025 is based on evidence from multiple studies conducted in China. Our study results primarily support the change in the vaccine administered to 6-year-old children from DT to co-purified DTaP.

The reported cases of pertussis have risen significantly in China over the last two years, with the majority of severe and fatal cases involving unvaccinated infants aged three months and younger. In addition, a clinical trial was undertaken in China to assess the immunogenicity and safety of co-purified DTaP vaccination at the ages of 2-4-6 months or 3-4-5 months. Based on this background, and to strengthen immune protection at earlier ages, the Chinese government has accelerated the timing of the first co-purified DTaP vaccine dose from 3 months to 2 months of age, adjusting the primary immunization series from 3-4-5 months to 2-4-6 months. To our knowledge, the clinical trial results comparing the immunogenicity and safety of the 2-4-6-month-old schedule and the 3-4-5-month-old schedule for the co-purified DTaP have not been publicly released yet, and this trial and our study are two independent ones. Therefore, we did not mention the research or references supporting the adjustment of the primary immunization series in the manuscript.

We have added background and reference on the adjustment of the co-purified DTaP immunization schedule in China in the Discussion section (Page 12 Line 317-324) as follows: In recent years, a substantial increase in pertussis incidence was observed in China, with a pronounced surge among children aged 5–9 years, while severe and fatal cases remain largely concentrated among unvaccinated infants aged ≤ 3 months. To enhance the immune protection for school-aged children and infants aged ≤ 3 months, China has implemented a new immunization schedule containing pertussis vaccine from January 1, 2025, which changes from one dose of co-purified DTaP vaccine at 3, 4, 5, and 18 months of age to one

dose at 2, 4, 6, 18 months, and 6 years of age⁴⁵.

Reference

45. Administration NDCAP. Notice Regarding Adjustments to the Immunization Schedule for DTaP and DT Vaccines under the National Immunization Program. https://www.ndcpa.gov.cn/jbkzxx/c100014/common/content/content_1872098276166717440.html (accessed 16 August 2025).

Additionally, the statement in the Discussion that “Our findings provided data on the immunogenicity and safety of co-purified DTaP vaccination in 6-year-old children, which support the adjustment of the DTaP immunization schedule in China” is not clearly supported by the data presented. Please elaborate on how the findings justify this recommendation and specify what kind of adjustment is being proposed (e.g., timing, number of doses, target population).

Response: Thank you very much for your constructive suggestions. We have revised the texts in the Discussion section (Page 12-13 Line 324-330) as follows: Our study found that in 6-year-old children, co-purified DTaP vaccination achieved a 70% seroconversion rate for pertussis anti-PT antibodies, significantly increased levels of pertussis anti-PT and anti-FHA antibodies, did not reduce diphtheria or tetanus antibody levels compared with DT vaccination, and demonstrated a good safety profile. These results support the strategy of administering a fifth dose of co-purified DTaP vaccine instead of DT vaccine to 6-year-old children in China's NIP.

Reviewer #2 (Remarks to the Author):

Wording here is still not quite clear to me:

"Response: Thank you for your comment. We have revised it in Page 7 Line 158-160 as follows: The incidence of \geq grade 3 vaccine-related adverse events was no significant difference between the two arms (0.83% vs. 0%, $P=0.499$)"

Response: We sincerely apologize for that. We have revised the texts in the Results section (Page 8 Line 180-182) as follows: There was no significant difference in the incidence of \geq grade 3 vaccine-related adverse events between the two arms (0.83% [2/240] vs. 0% [0/240], $P=0.499$).

Reviewer #3 (Remarks to the Author):

Reviewer #3

4: The authors' response to this comment is inappropriate. When comparing rates between two groups, we need to specify the target rates of both groups for a sample size calculation. That is, we cannot calculate a sample size for a chi-square test (or Fisher exact test) without specifying the expected (or target) rates of both groups.

Response: Thank you for your additional feedback. We recognize the statistical issues in our manuscript. Regarding pertussis anti-PT and anti-FHA antibodies, we have discontinued using the χ^2 test for between-arm comparisons of post-vaccination

seropositivity rates, seroconversion rates, and GMCs. Instead, we have implemented a single-arm objective performance criteria for anti-PT antibody seroconversion rates against a 70% target value, where the criterion for success requires the lower limit of the 95% CI for the anti-PT seroconversion rate to exceed 70%. For anti-DT and anti-TT antibodies, we continue to use the χ^2 test for between-arm comparisons of post-vaccination seropositivity rates, seroconversion rates, and GMCs. Relevant content has been revised in the Methods, Results, Discussion, and Tables sections.

Page 16-17, Line 440-443, Methods:

As PT is a key protective antigen in pertussis vaccines⁴⁶, the seroconversion rate for anti-PT antibodies was used for sample size estimation. The sample size was estimated using an objective performance criteria method. Assuming the target value was 70%³⁹ of the seroconversion rate of anti-PT in the co-purified DTaP after vaccination, with a two-sided α of 0.05, a power of 90%, and a potential loss to follow-up of 15%, we estimated at least a sample size of 240 participants per arm.

Reference

39. Zhu F, et al. Booster vaccination against pertussis in Chinese children at six years of age using reduced antigen content diphtheria-tetanus-acellular pertussis vaccine (Boostrix()). *Human vaccines* 6, (2010).

46. Gregg KA, Merkel TJ. Pertussis Toxin: A Key Component in Pertussis Vaccines? *Toxins* 11, (2019).

Page 17, Line 449-457, Methods:

A single-arm objective performance criteria for anti-PT antibody seroconversion rates against a 70% target value, where the criterion for success requires the lower limit of the 95% CI for the anti-PT seroconversion rate to exceed 70%. We used the χ^2 test to compare seropositive rates and seroconversion rates of anti-DT and anti-TT between arms. The Wilson (score) method was used to calculate 95% confidence intervals (CIs) for seropositive rates and seroconversion rates. After \log_{10} transformation of antibody concentrations, GMCs of anti-DT and anti-TT antibodies were compared between arms using the Student's t-test.

Page 6, Line 147-157, Results:

On 28 days after vaccination, the seropositive rates of anti-pertussis antibodies in the co-purified DTaP increased to 85.47% (200/234, 95% CI: 80.38-89.41) for PT and 97.44% (228/234, 95% CI: 94.52-98.82) for FHA. The seroconversion rate for anti-PT antibody was 81.20% (190/234, 95% CI: 75.70-85.68), with the lower limit of the 95% CI exceeding 70% (75.70% > 70%). The seroconversion rate for anti-FHA antibody was 74.36% (174/234, 95% CI: 68.40-79.53).

Page 7, Line 162-168, Results:

The anti-DT, and anti-TT had higher GMCs or median concentrations in the co-purified DTaP than in the DT (Table 2, Figure 2). Participants with negative pertussis antibodies at baseline had a higher seroconversion rate than those with positive pertussis antibodies at baseline in the co-purified DTaP (Supplemental Table 1).

Page 8, Line 196-200, Discussion:

In our trial conducted in Zhejiang province, we found that after 6-year-old children were vaccinated with the co-purified DTaP, seroconversion rate of anti-PT reached 81.20% (target value:70%), and both seropositive rates and antibody GMCs against pertussis (anti-PT and anti-FHA) increased significantly.

Reviewer #1 (Remarks to the Author):

While the authors have addressed previous concerns, several pivotal issues require further elaboration to fully solidify the manuscript's arguments and ensure its conclusions are robustly supported.

Major Comments:

- The accurate interpretation of immunogenicity results is contingent upon the baseline health status of participants. The reliance on guardian-reported "no history of pertussis" as a sole inclusion criterion represents a significant methodological limitation, especially given the high community transmission and potential for asymptomatic or mild infections. The authors must clarify:

The specific case definition used to screen and exclude potential participants. Was active surveillance for pertussis-like symptoms conducted in participants or their close contacts prior to enrollment?

Whether the local health system's standardized case definition (incorporating laboratory confirmation, clinical criteria, and/or epidemiological linkage) was applied during the screening process.

The potential for undiagnosed pre-enrollment infection to confound the immunogenicity results—for instance, by inflating the apparent vaccine-induced response in a subset of participants—must be explicitly acknowledged and discussed as a key study limitation.

Response: Thank you for your valuable suggestion and reminder. We acknowledge that relying on guardian-reported "no history of pertussis" as the primary criterion for screening pertussis infection represents a significant methodological limitation.

In our study, active surveillance for pertussis-like symptoms was not conducted in participants or their close contacts prior to enrollment, and participants were not screened for pertussis infection based on a standardized case definition (such as laboratory confirmation, clinical criteria, and/or epidemiological links) before vaccination. As a result, some participants with asymptomatic or moderate symptomatic pertussis infection who did not visit health care facilities may have been recruited in the trial, which could have confounded the immunogenicity results.

We have added the relevant content in the Discussion section (Page 11 Line 291-298) as follows: First, although guardians were asked to self-report a history of pertussis at enrollment, active surveillance for pertussis-like symptoms was not conducted in participants or their close contacts prior to enrollment, and participants were not screened for pertussis infection based on a standardized case definition (such as laboratory confirmation, clinical criteria, and/or epidemiological links) before vaccination. As a result, some participants with asymptomatic or moderate symptomatic pertussis infection who did not visit health care facilities may have been recruited in the trial, which could have confounded the immunogenicity results.

- The observed lower seroconversion rates among participants who were seropositive at baseline is a critical finding. This topic needs a more profound discussion in the context of current immunological understanding.

For a better interpretation of the results, the authors should explicitly state that, for pertussis, a

universally accepted serological correlate of protection remains elusive. While the anti-PT seroconversion rate is encouraging and meets the pre-specified target, the discussion must clearly articulate that the direct translation of these antibody levels into clinical efficacy and durable protection is uncertain.

Response: Thank you for the reminder. We have added relevant content about the impact of baseline antibody levels on vaccine responses in the Discussion section (Page 9 Line 246-254) as follows: Some studies on Tdap booster immunization have shown that participants with high baseline pertussis antibody levels have a higher antibody response one month after vaccination, suggesting that the long-lasting memory formed by previous exposure or vaccination history may affect the vaccine response⁴³. However, participants who were positive for pertussis antibodies at baseline had lower seroconversion rates of PT and FHA after a booster vaccination with co-purified DTaP than those who were negative for pertussis antibodies at baseline in our trial, which could be attributed to the different definitions of seroconversion rates for pre-immunization positive or negative.

References

43. Knuutila A, et al. Pertussis toxin neutralizing antibody response after an acellular booster vaccination in Dutch and Finnish participants of different age groups. *Emerging microbes & infections* 11, 956-963 (2022).

We have added relevant content about the translation of pertussis antibody levels into clinical efficacy and durable protection in the Discussion section (Page 8 Line 196-203) as follows: The correlation between pertussis antibody thresholds and efficacy and durable protection remains uncertain³⁷. Although the seroconversion rate against PT in our trial met the pre-specified target, it is necessary to evaluate the effectiveness of the co-purified DTaP vaccine against the current circulating pertussis strains.

References

37. Kapil P, Merkel TJ. Pertussis vaccines and protective immunity. *Current opinion in immunology* 59, 72-78 (2019).

- The "co-purified" nature of the vaccine is a central differentiator, yet its implications for the final product's composition and contemporary relevance require further clarification. Beyond the stipulation that PT and FHA constitute $\geq 85\%$ of the protein content, the authors should elaborate on the confirmed identity and batch-to-batch consistency of the remaining antigenic components (e.g., PRN, FIM2/3).

Response: Thank you for your valuable suggestion. The Chinese Pharmacopoeia stipulates that for co-purified DTaP vaccines, the content of PT and FHA shall constitute no less than 85% of the total protein content, while no relevant requirements are set for the remaining antigenic components (e.g., PRN, FIM2/3). Therefore, neither the vaccine manufacturers nor the National Institutes of Food and Drug Control have included the detection of PRN and FIM 2/3 antigen component content and batch-to-batch consistency in their routine production and vaccine lot release work.

There is little literature on the study of the content of PRN and FIM 2/3 antigen components in co-purified DTaP vaccine. Limited research evidence indicates that the co-purified DTaP contains PRN and FIM 2/3 antigen components, batch-to-batch

consistency in terms of the protein amounts was stable, while those from manufacturers were varied significantly. We suppose that there is still a lack of sufficient research evidence to illustrate the proportion and batch-to-batch consistency of the remaining antigen components^{1,2}. Therefore, we did not further elaborate on the confirmation of the identity and batch-to-batch consistency of other antigen components (such as PRN and FIM 2/3) in the manuscript.

References

1. Long Z, et al. Quantitative determination of bioactive proteins in diphtheria tetanus acellular pertussis (DTaP) vaccine by liquid chromatography tandem mass spectrometry. *Journal of pharmaceutical and biomedical analysis* 169, 30-40 (2019).
2. Xu Y, Tan Y, Asokanathan C, Zhang S, Xing D, Wang J. Characterization of co-purified acellular pertussis vaccines. *Human vaccines & immunotherapeutics* 11, 421-427 (2015).

- Furthermore, to contextualize the vaccine's current relevance, information on the genetic characteristics of the production strain (*B. pertussis* Bost strain) and a comparison of its antigenic profile with currently circulating strains in China (e.g., the prevalent ptxP3 genotype) would be highly valuable. A more detailed characterization in the Discussion would facilitate a more meaningful comparison with component-purified vaccines and provide deeper insight into the immune responses observed.

Response: Thank you for your suggestion. We have added the relevant content in the Discussion section (Page 7 Line 190-196) as follows: Similar to many industrialized countries with high vaccination coverage, China has also reported a shift in the circulating strains of *Bordetella pertussis* in recent years^{32, 33, 34, 35}. A significant divergence between circulating strains ptxP3 and co-purified DTaP vaccine strain (CS strain: ptxP-1/prn-1/ptxA-2/ffm2-1/ffm3-1/tcfA2) was observed in China^{35, 36}. The shift of ptxP3 variants from ptxP1 vaccine strains may facilitate vaccine-elicited immunity escape, which could partially explain the resurgence of pertussis.

References

32. Clarke M, et al. The relationship between *Bordetella pertussis* genotype and clinical severity in Australian children with pertussis. *The Journal of infection* 72, 171-178 (2016).
33. Brandal LT, et al. Evolution of *Bordetella pertussis* in the acellular vaccine era in Norway, 1996 to 2019. *European journal of clinical microbiology & infectious diseases* : official publication of the European Society of Clinical Microbiology 41, 913-924 (2022).
34. Bart MJ, et al. Global population structure and evolution of *Bordetella pertussis* and their relationship with vaccination. *mBio* 5, e01074 (2014).
35. Cai J, et al. Waning immunity, prevailing non-vaccine type ptxP3 and macrolide-resistant strains in the 2024 pertussis outbreak in China: a multicentre cross-sectional descriptive study. *The Lancet regional health Western Pacific* 60, 101628 (2025).
36. Li Z, et al. Genomic epidemiology and evolution of *Bordetella pertussis* under the vaccination pressure of acellular vaccines in Beijing, China, 2020-2023. *Emerging microbes & infections* 14, 2447611 (2025).

- To further strengthen the safety discussion, the authors should contextualize the reactogenicity findings by providing a direct comparison of adverse event rates (particularly local reactions) with

those reported for internationally used Tdap or DTaP booster vaccines in similar age groups. Such a comparison is essential to determine whether the observed reactogenicity profile is unique to this vaccine platform or consistent with acellular pertussis boosters in general.

Response: Thank you very much for your constructive suggestions. We have added the relevant content in the Discussion section (Page 10 Line 266-271) as follows: In our trial, more than 15% of the participants who received co-purified DTaP experienced redness or swelling. A large number of studies in the United States, Canada and other countries have found that local reactions are the most common adverse events after Tdap or component DTaP booster vaccination in similar age groups^{44, 45}, which is consistent with the safety results of our study.

References

44. Langley JM, et al. An adolescent-adult formulation tetanus and diphtheria toxoids adsorbed combined with acellular pertussis vaccine has comparable immunogenicity but less reactogenicity in children 4-6 years of age than a pediatric formulation acellular pertussis vaccine and diphtheria and tetanus toxoids adsorbed combined with inactivated poliomyelitis vaccine. *Vaccine* 25, 1121-1125 (2007).

45. Use of diphtheria toxoid-tetanus toxoid-acellular pertussis vaccine as a five-dose series. Supplemental recommendations of the Advisory Committee on Immunization Practices (ACIP). *MMWR Recommendations and reports : Morbidity and mortality weekly report Recommendations and reports* 49, 1-8 (2000).

- The limitations section should be refined for greater precision. The constraint of a single-province study site should be framed not merely as a geographical limitation, but as a potential constraint on the generalizability of the findings to populations with differing background rates of pertussis circulation, natural immunity, and potentially heterogeneous circulating *B. pertussis* strains.

Response: Thank you for your suggestion. We have added the relevant content in the Discussion section (Page 11 Line 299-304) as follows: Second, we conducted our trial in Zhejiang Province, which has a high reporting rate of pertussis cases in China. This might impose potential constraint on the generalizability of the findings to populations with differing background rates of pertussis circulation, natural immunity, and potentially heterogeneous circulating *Bordetella pertussis* strains.